# Formation and three-dimensional architecture of *Leishmania* adhesion in the sand fly vector

**Ryuji Yanase[1], Flávia Moreira-Leite[1], Edward Rea[1], Lauren Wilburn[1], Jovana Sádlová[2], Barbora Vojtkova[2], Katerina Pružinová[2], Atsushi Taniguchi[3,4], Shigenori Nonaka[4,5,6], Petr Volf[2], Jack D Sunter[1]***

[1]Department of Biological and Medical Sciences, Oxford Brookes University, Oxford, United Kingdom; [2]Department of Parasitology, Charles University, Prague, Czech Republic; [3]Research Center of Mathematics for Social Creativity, Research Institute for Electronic Science, Hokkaido University, Sapporo, Japan; [4]Laboratory for Spatiotemporal Regulations, National Institute for Basic Biology, Okazaki, Japan; [5]Spatiotemporal Regulations Group, Exploratory Research Center for Life and Living Systems, Okazaki, Japan; [6]Department of Basic Biology, School of Life Science, Okazaki, Japan

**Abstract** Attachment to a substrate to maintain position in a specific ecological niche is a common strategy across biology, especially for eukaryotic parasites. During development in the sand fly vector, the eukaryotic parasite *Leishmania* adheres to the stomodeal valve, as the specialised haptomonad form. Dissection of haptomonad adhesion is a critical step for understanding the complete life cycle of *Leishmania*. Nevertheless, haptomonad studies are limited, as this is a technically challenging life cycle form to investigate. Here, we have combined three-dimensional electron microscopy approaches, including serial block face scanning electron microscopy (SBFSEM) and serial tomography to dissect the organisation and architecture of haptomonads in the sand fly. We showed that the attachment plaque contains distinct structural elements. Using time-lapse light microscopy of in vitro haptomonad-like cells, we identified five stages of haptomonad-like cell differentiation, and showed that calcium is necessary for *Leishmania* adhesion to the surface in vitro. This study provides the structural and regulatory foundations of *Leishmania* adhesion, which are critical for a holistic understanding of the *Leishmania* life cycle.

*For correspondence:
jsunter@brookes.ac.uk

Competing interest: The authors declare that no competing interests exist.

## Editor's evaluation

This important work substantially advances our understanding of the organization and architecture of haptomonads, a distinct and poorly understood developmental form of Leishmania found in sand fly vectors at later stages of infection. The 3D electron microscopy methodology, including serial block face scanning electron microscopy and electron tomography to visualize haptomonads in sand fly, is exceptional and establishes a new standard in the field.

## Introduction

Attachment to a substrate to maintain position in a specific permissive ecological niche is a commonly exploited strategy across biology. It has especially been employed as a strategy for escape from host defences and for vector transmission to the host by pathogens including many eukaryotic unicellular parasites such as *Plasmodium*, *Giardia* and the kinetoplastids, including *Leishmania* spp., *Trypanosoma*

*cruzi*, and *Trypanosoma congolense* (*Beattie and Gull, 1997*; *Dvorak et al., 1975*; *Evans et al., 1979*; *Friend, 1966*; *Killick-Kendrick et al., 1974*; *Tetley and Vickerman, 1985*; *Bloodgood, 1990*). For example, *Plasmodium falciparum* constructs knobs for attachment to host tissues to avoid clearance. Knobs are organised multi-protein structural complexes with specificity defined by the PfEMP1 variant expressed (*Jensen et al., 2020*).

*Leishmania* spp. are flagellated eukaryotic parasites that cause leishmaniasis, a neglected tropical disease with a range of different pathologies (*Burza et al., 2018*). *Leishmania* has a complex life cycle with multiple developmental forms as it cycles between a sand fly vector and a mammalian host (*Sunter and Gull, 2017*). During parasite development in the sand fly, the parasite adheres to and colonises the stomodeal valve at the anterior end of the midgut (*Figure 1A*; *Dostálová and Volf, 2012*). The attached form is called the haptomonad and is characterised by a reduced flagellum with an enlarged flagellar tip that contains a complex and poorly characterised set of cytoskeletal structures (*Killick-Kendrick et al., 1974*; *Killick-Kendrick et al., 1977*; *Molyneux et al., 1975*). These structures including the attachment plaque at the membrane-substrate interface form a strong connection between the parasite and the underlying cuticle of the stomodeal valve. The role of the haptomonad form is not fully understood but it is likely required to maintain a persistent infection in the sand fly vector, and contributes to the destruction and obstruction of the stomodeal valve by the secretion of chitinase and the formation of the gel-like plug and the haptomonad parasite sphere. These facilitate reflux and the transmission of parasites during the feeding on the vertebrate host (*Bates, 2007*; *Hall et al., 2021*; *Rogers et al., 2008*; *Rogers et al., 2002*; *Serafim et al., 2018*; *Volf et al., 2004*).

In transmission electron microscopy (TEM) images, the haptomonad form attachment plaque is strikingly reminiscent of the hemidesmosomes that attach epithelial cells to the underlying extracellular matrix. Hemidesmosomes are formed by a set of transmembrane proteins (integrins and BP180), which connect the extracellular basal lamina to intracellular adaptor proteins (Plectin and BP230) that bind to intermediate filaments (*Borradori and Sonnenberg, 1999*). The assembly mechanism of the related desmosome, which connects two adjacent cells, requires calcium; the removal of calcium results in the detachment of connected cells (*Garrod and Chidgey, 2008*). However, desmosomes are able to reversibly switch to a hyper-adhesive state, which is more stable and remains assembled even when calcium is removed (*Garrod et al., 2005*). Previous studies suggest that trypanosomatid attachment in vitro is resistant to divalent cation depletion and it is strengthened, rather than weakened, by the removal of divalent cations (*Hendry, 1987*; *Molyneux et al., 1987*; *Bloodgood, 1990*). However, no studies have examined in detail the involvement of calcium for adhesion in *Leishmania*.

There are few studies of the haptomonad form, as this is a technically challenging life cycle form to investigate and thus the most 'neglected' (*Cecílio et al., 2022*). Previous haptomonad analysis has been restricted to thin-section TEM of *L. mexicana* infected sand flies and in-vitro-derived haptomonad-like cells attached to surfaces, with haptomonads being defined by their morphology and the formation of an attachment plaque (*Killick-Kendrick et al., 1977*; *Killick-Kendrick et al., 1974*; *Molyneux et al., 1975*; *Wakid and Bates, 2004*). In this study, we refer to in vitro generated attached cells that morphologically resemble in vivo haptomonads, have a short and wide cell body, and attaches to substrates through the flagellum with an attachment plaque as in vitro haptomonad-like cells. While those 2D TEM images in the previous studies afforded a basic description of the cytoskeletal architecture of the attachment plaque, they provided limited information on the 3D architecture of the flagellar pocket and flagellum at the attachment region, and on the relationship between individual haptomonads in the stomodeal valve.

Here, we have combined powerful 3D electron microscopy (volume EM - vEM) approaches to image haptomonad forms in the sand fly, enabling us to synthesise a unified view of *Leishmania* adhesion in the vector. Our data allowed us to describe the intracellular attachment architecture in unprecedented detail, while placing this architecture in the context of spatial organisation of haptomonad populations. Using time-lapse microscopy, we observed the adhesion process of in vitro haptomonad-like cells and determined that in vitro *Leishmania* adhesion occurs through a defined series of steps. Finally, we showed that calcium is necessary for in vitro adhesion. Overall, this defines the structural and regulatory foundations of *Leishmania* adhesion, which are critical for a holistic understanding of *Leishmania* life cycle.

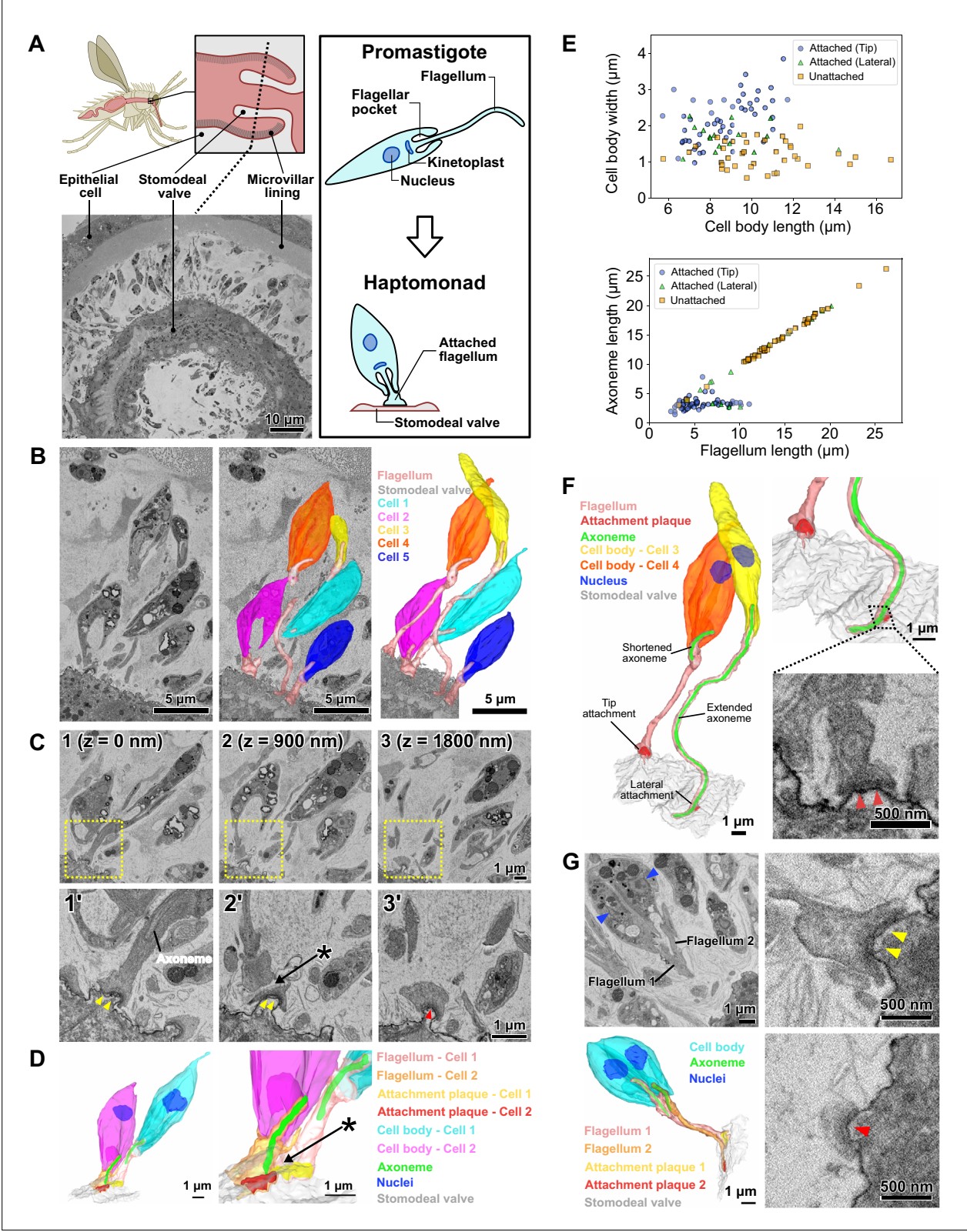

**Figure 1.** Haptomonads form a dense, multi-layered complex on the sand fly stomodeal valve. (**A**) Cartoon of the sand fly digestive tract highlighting the position of the stomodeal valve, the site of differentiation of free-swimming *Leishmania* promastigotes into haptomonads, which attach to the stomodeal valve surface via a shortened and expanded flagellum (Right black box). SBFSEM slice (bottom left) showing a cross section of the stomodeal valve at the position indicated by the dotted line in the cartoon (***Figure 1—video 1***). (**B**) SBFSEM slice and 3D reconstruction showing attached

*Figure 1 continued on next page*

*Figure 1 continued*

haptomonads in close proximity to each other and attached to the stomodeal valve (grey) by flagella (peach) of different lengths (*Figure 1—videos 2–3*). (C) Sequential SBFSEM images (every 900 nm) showing details of the attached flagella from cells 1 and 2 in C. Top: lower magnification view showing entire cells. Bottom: higher magnification view of the attachment region indicated by the yellow dotted box in the top image. Both cells were attached to the stomodeal valve by a flagellum containing an attachment plaque (yellow and red arrowheads), but no attachment plaque was formed between neighbouring flagella (asterisk). (D) 3D reconstruction of the inner flagellum structures from cells 1 and 2 in C, showing the attachment plaques and the lack of attachment between flagella (asterisk, as indicated in D). (E) Scatter plots of cell body length against cell body width (Top), and flagellum length against axoneme length (Bottom) of attached cells with the distal flagellar tip attachment (Blue circles; n=50), attached cells with the laterally attached flagellum (Green triangles; n=17), and unattached cells (Orange squares; n=39) in the sand fly. (F) 3D reconstruction (left and top right) showing a slender and wider cell (cell 3 and 4 respectively in C) attached to the stomodeal valve via an attachment plaque formed on the lateral part or the distal tip of the flagellum, respectively. Note: the axoneme extended along the length of the flagellum in cell 3 with the laterally attached flagellum, whereas in the cell 4 with the distal flagellar tip attachment, the axoneme only just extended beyond the end of the cell body. SBFSEM image (bottom right) showing a cross section of an attachment plaque (red arrowheads) formed on the lateral part of the flagellum of cell 3. (G) SBFSEM slices (top left and right) and 3D reconstruction (bottom left) showing a dividing haptomonad with two nuclei (blue arrowheads) and two attached flagella. Top and bottom right: higher magnification views of the attachment plaques in each flagellum (yellow and red arrowheads, corresponding to the colours of the attachment plaques in the 3D reconstruction).

The online version of this article includes the following video and figure supplement(s) for figure 1:

**Figure supplement 1.** Cell cycle analysis of haptomonads at the sand fly stomodeal valve.

**Figure supplement 2.** Haptomonads detached from the stomodeal valve.

**Figure supplement 3.** Damage to the stomodeal valve at points of haptomonad attachment.

**Figure 1—video 1.** SBFSEM imaging of haptomonads on the stomodeal valve in the sand fly (whole view).

https://elifesciences.org/articles/84552/figures#fig1video1

**Figure 1—video 2.** SBFSEM imaging of haptomonads on the stomodeal valve in the sand fly (enlarged view).

https://elifesciences.org/articles/84552/figures#fig1video2

**Figure 1—video 3.** 3D model generated from the SBFSEM imaging shown in *Figure 1—video 2*.

https://elifesciences.org/articles/84552/figures#fig1video3

**Figure 1—video 4.** 3D model of mapping of an infected stomodeal valve showing the position of the nuclei of different attached cells.

https://elifesciences.org/articles/84552/figures#fig1video4

# Results

## Dense and complex organisation of haptomonads on the sand fly stomodeal valve

To understand the spatial organisation and architecture of *L. mexicana* haptomonads colonising the stomodeal valve, we used serial block face scanning electron microscopy (SBFSEM) to examine fixed and dissected midguts from sand flies that had been infected with *L. mexicana* for 10 days (*Figure 1—video 1*; *Figure 1*). At this stage, there was a dense infection around the stomodeal valve, with many haptomonads attached to the cuticle surface (*Figure 1—video 2*; *Figure 1A–B*). Around the stomodeal valve, the haptomonads were present on both sides, in a series of layers relative to the cuticle surface, with some cells very close to the surface and others further away; the flagellum length varied according to the distance to the surface (*Figure 1—video 3*; *Figure 1B*). In SBFSEM images, the attachment plaque within the flagellum appeared as an electron-dense layer overlaying the flagellar membrane and the flagellum of haptomonads attached exclusively to the cuticle surface, and no connections were observed between adjacent cells or flagella (*Figure 1C–D*). Attached cells had a shorter and wider cell body (Average length = 8.7 ± 1.6 µm (s.d.); average width = 2.2 ± 0.6 µm (s.d.), n=50) compared to unattached cells (Average length = 10.5 ± 2.6 µm (s.d.); average width = 1.2 ± 0.4 µm (s.d.), n=39; *Figure 1E*). The majority (83/100) of cells were attached via a connection at the distal tip of the flagellum, with the rest (17/100) connected laterally through the side of the flagellum (*Figure 1F*). In unattached cells, the microtubule axoneme extended to the tip of the flagellum (Average axoneme length = 14.2 ± 4.6 µm (s.d.), n=39), whereas in attached cells with a flagellar tip attachment, the axoneme only just extended beyond the end of the cell body (Average axoneme length = 3.3 ± 1.0 µm (s.d.), n=50), regardless of the length of the flagellum (*Figure 1D–F*). Those attached cells with a relatively long flagellum and shortened axoneme may have completed differentiation into the haptomonad form, but due to the high cell density at the stomodeal valve, there may be insufficient space to allow them to fully disassemble their flagellum (*Figure 1B and F*).

In the cells with the laterally attached flagellum, the axoneme extended further along the flagellum (Average axoneme length = 10.0 ± 6.2 μm (s.d.), n=17) and their cell body length and width were intermediate between attached cells with a flagellar tip attachment and unattached cells (Average length = 9.1 ± 1.9 μm (s.d.); average width = 1.6 ± 0.4 μm (s.d.), n=17), suggesting that these parasites may be in the process of differentiating to or away from the haptomonad form (*Figure 1E–F*). Most of unattached cells and cells with a laterally attached flagella had a metacyclic/leptomonad morphology, which is consistent with previous reports of the cell type in the sand fly cardia at late stages of infection (*Figure 1E*; *Rogers et al., 2002*).

We took advantage of the large numbers of cells visualised by our SBFSEM approach to determine the replication status of the cells. We examined over 1000 cells from two different stomodeal valves and defined those cells with either one or two nuclei and two dividing flagella as dividing (1N2F or 2N2F; *Figure 1—video 4*; *Figure 1—figure supplement 1*). The vast majority of haptomonads were non-dividing with one nucleus and one flagellum (1N1F), matching previous reports *Gossage et al., 2003*; however, we identified a small number of dividing (1N2F or 2N2F) haptomonad forms (*Figure 1—figure supplement 1*). In those dividing cells with a long new flagellum, both the old and the new flagella were attached to the cuticle (*Figure 1G*), but in dividing cells with a short new flagellum only the old flagellum was attached. In addition to the dividing and non-dividing cells, we identified four cells with either three nuclei or two nuclei and only one flagellum (≥2N1 F), which are likely abnormal, though they may represent rare examples of cell fusion events or gamete production (*Figure 1—figure supplement 1*). Also, we found a small number of haptomonads (6 cells in ~1000 cells) with a flagellum attachment plaque that had partially or completely detached from the stomodeal valve. While it is possible that these cells became detached from the stomodeal valve during tissue fixation, these partially detached forms may also represent haptomonads in the process of differentiating back into a free-swimming life cycle form, to continue the life cycle (*Figure 1—figure supplement 2*).

In addition, as has been observed in previous studies (*Rogers et al., 2008*; *Schlein et al., 1992*; *Volf et al., 2004*), we observed a detachment of the cuticular lining from the epithelial cells at the areas of the stomodeal valve where a large number of haptomonads were adhered (*Figure 1—figure supplement 3*).

## The haptomonad flagellum is a highly modified and specialised organelle

To obtain more detailed 3D information on the attachment structure of individual haptomonads, we stopped SBFSEM imaging half-way through the stomodeal valve area where haptomonad cells of comparable density to those observed in the heavily infected stomodeal valve of a previous study were seen (*Rogers et al., 2008*), and then semi-thin (~150 nm) serial sections were cut from the SBFSEM sample block face, for use in serial electron tomography (*Figure 2—video 1*). A 3D model was created from the reconstructed serial tomogram, and the detailed 3D configuration of the attached flagellum and the region around the flagellar pocket of the haptomonad cell was examined (*Figure 2—video 2*; *Figure 2A*). As observed in the SBFSEM data, the haptomonad was attached to the stomodeal valve through the distal tip of the shortened flagellum, and the attachment interface was covered with an electron-dense attachment plaque (*Figure 2A–B*). The anterior tip of the cell body in promastigotes is asymmetrical with the cell body extending along the side of the flagellum in which the flagellum attachment zone (FAZ) is found, with its typical electron-dense junctional complexes (*Sunter et al., 2019*; *Wheeler et al., 2016*). In the haptomonad cell, this asymmetric cell body extension is much shorter, with a wider anterior cell tip (*Figure 2A–C*). The haptomonad flagellar pocket consisted of a bulbous region at its base and a neck region more closely apposed to the flagellum, with the microtubule quartet running over the bulbous region and into the neck region. In comparison with the promastigote, the neck region was shorter, and as the flagellum exited the neck region there was an expansion of the flagellum, with a large number of junctional complexes connecting the flagellar membrane to the cell body membrane (*Figure 2A and C*). A 9+2 microtubule axoneme was present within the flagellum; however, the axoneme extended only just beyond the cell body and the central pair microtubules were not present in the final few hundred micrometres (*Figure 2D–F*). Markham rotational averaging (*Gadelha et al., 2006*; *Skalický et al., 2017*) showed that the 9+2 axoneme was associated with accessory structures required for motility including radial spokes and inner and outer

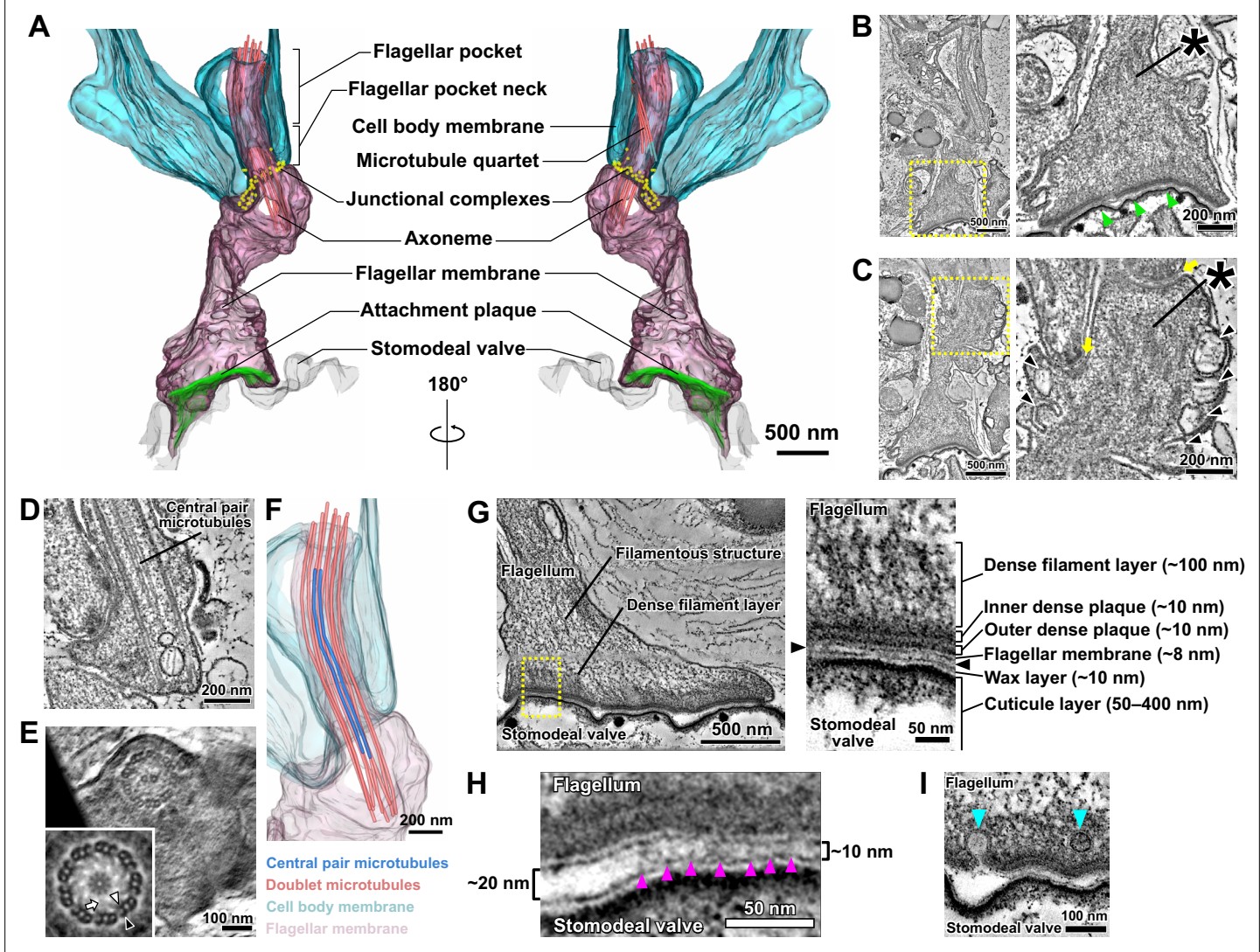

**Figure 2.** The modified haptomonad flagellum has a highly organised ultrastructure, with multiple discrete elements. Different serial tomograms of haptomonads in the sand fly, focusing on the attachment region (*Figure 2—video 1*). (**A–C**) 3D reconstruction (*Figure 2—video 2*; **A**) and slices (**B**, **C**) from the serial tomogram shown in *Figure 2—video 1*. (**B**) Low- (left) and high- (right) magnification tomogram slices showing the attachment plaque region (green arrowheads), the expanded flagellum attachment zone (FAZ; yellow arrows) found in attached cells, as well as filament bundles (asterisks) and vesicles (black arrowheads) found inside the attached flagellum. (**D, E**) Slices from a different tomogram showing the shortened haptomonad 9+2 axoneme, in longitudinal section (**D**) and cross-section (**E**). The nine-fold Markham rotational averaging of the same axoneme (inset in E) highlights the presence of inner and outer dynein arms (white and black arrowheads, respectively) and radial spokes (white arrow). (**F**) 3D reconstruction of the shortened 9+2 axoneme shown in D. (**G–I**) Slices from a higher magnification tomogram of the attachment region. The image on the right in G shows a high-magnification view of the boxed area on the left image, with the distinct ultrastructural elements of the attachment region indicated. The black arrowheads indicate the gap between the flagellum and the stomodeal valve. (**H**) Magnified view of the gap between the flagellum and the stomodeal valve in a different area of the tomogram. Electron-dense structures (magenta arrowheads) appear to connect the flagellar membrane to the stomodeal valve surface in the narrower (~10 nm wide) regions of the gap, but were absent in the wider (~20 nm wide) region of the gap. (**I**) Different area of the tomogram showing vesicles (cyan arrowheads) budding from (or fusing with) the flagellar membrane at the attachment interface.

The online version of this article includes the following video and figure supplement(s) for figure 2:

**Figure supplement 1.** Tip structure of the in vivo haptomonad and in vitro haptomonad-like cell axoneme.

**Figure 2—video 1.** Serial section electron tomography of a haptomonad on the stomodeal valve in the sand fly.
https://elifesciences.org/articles/84552/figures#fig2video1

**Figure 2—video 2.** 3D model generated from the serial tomogram shown in *Figure 2—video 1*.
https://elifesciences.org/articles/84552/figures#fig2video2

dynein arms (*Figure 2E*). In addition, the tip of the shortened outer doublet microtubules and central pair microtubules were capped with amorphous material and a ring-like capping structure (*Figure 2—figure supplement 1*). Finally, in the haptomonad flagellum, we did not observe the paraflagellar rod (PFR), an extra-axonemal structure (*Figure 2D–F*; *Portman and Gull, 2010*).

Filaments and filament bundles were present throughout the attached flagellum extending from the FAZ to the attachment interface over a distance of ~3 µm (*Figure 2B–C and G*). These filamentous structures appeared to connect the attachment interface to the cell body via the junctional complexes (*Figure 2C*). A complex of attachment plaque structures was present at the attachment interface of the flagellar membrane connected to the surface of the stomodeal valve (*Figure 2G*). The attachment plaque complex consisted of several structural layers, with outer and inner dense plaques, each approximately 10 nm thick, just inside the flagellar membrane, with a dense filamentous layer (~100 nm thick) emerging from the inner dense plaque and extending into the filamentous structures that ran towards the FAZ (*Figure 2G*). We also found that the stomodeal valve had a layered structure (*Figure 2G*). A thin layer similar in appearance to the flagellar membrane, approximately 10 nm thick, was observed overlaying the cuticular layer. This is likely the wax layer that is composed of lipidic components and is analogous to insect epicuticle (*Figure 2G*; *Schmidt et al., 1998*; *Vincent, 2001*). No readily observable differences in the superficial and inner structure of the stomodeal valve were found between regions where haptomonads were attached, compared with the region lacking attached parasites.

Across the majority of the attachment interface the flagellar membrane was positioned very close (~10 nm) to the stomodeal valve surface and connecting structures spanning this gap between the flagellar membrane and the putative wax layer of the stomodeal valve were seen (*Figure 2G–H*). Within the attachment interface, there were also gaps where the flagellar membrane was further away from the stomodeal valve surface; these gaps corresponded to regions where these connecting structures were absent (*Figure 2H*).

The haptomonad flagellar membrane did not encase the flagellum smoothly, and there were numerous projections and indentations within the membrane (*Figure 2B–C*). Furthermore, large vesicular structures were observed throughout the flagellum (*Figure 2C*). Intriguingly, several smaller vesicles were seen near the attachment interface, and in one instance, a vesicle appeared to be fusing with or budding from the flagellar membrane (*Figure 2I*).

## In vitro generated haptomonad-like cells morphologically resemble sand fly haptomonads

In vitro haptomonad-like cells have previously been generated by allowing promastigote cells to adhere to scratched plastic (*Maraghi et al., 1987*; *Wakid and Bates, 2004*). We used this approach here to generate in vitro haptomonad-like cells, which allowed us to analyse aspects of the adhesion process that are difficult to study in vivo, such as initial adhesion formation and dynamics. In scanning electron microscopy (SEM) imaging of cultures 72 hr after seeding promastigote cells onto the scratched plastic coverslips, we observed large clumps of cells that contained a mixture of haptomonad-like cells with a short flagellum attached to the substrate via an attachment plaque and promastigote cells with an unattached long flagellum, in addition to individual haptomonad-like cells with a short attached flagellum (*Figure 3A*). We examined the organisation and architecture of the in vitro generated haptomonad-like cells by SBFSEM (*Figure 3—video 1*; *Figure 3B*). In a reassembled and modelled volume through an attached cell clump, we observed haptomonad-like cells clearly attached to the substrate with an organisation that resembled the haptomonads in the sand fly. The cell body was short and wide with a short flagellum that was attached through its tip to the substrate, with an electron-dense region at the attachment interface (*Figure 3B*). In addition to unattached cells within the clump, many more dividing (1N2F or 2N2F) cells were seen than in the sand fly (*Figure 3—video 2*; *Figure 3—figure supplement 1*). The unattached cells appeared to be trapped by a filamentous extracellular matrix, which was also found between haptomonad cells in the sand fly (*Figure 3—figure supplement 2*).

We used serial section electron tomography to examine the 3D ultrastructure of the in vitro haptomonad-like cells. The overall organisation of the attachment structure of the in vitro haptomonad-like cell was highly similar to that of the sand fly haptomonads. The in vitro cells had a shortened attached flagellum with a 9+2 microtubule axoneme, without the PFR, a shorter and a wider anterior

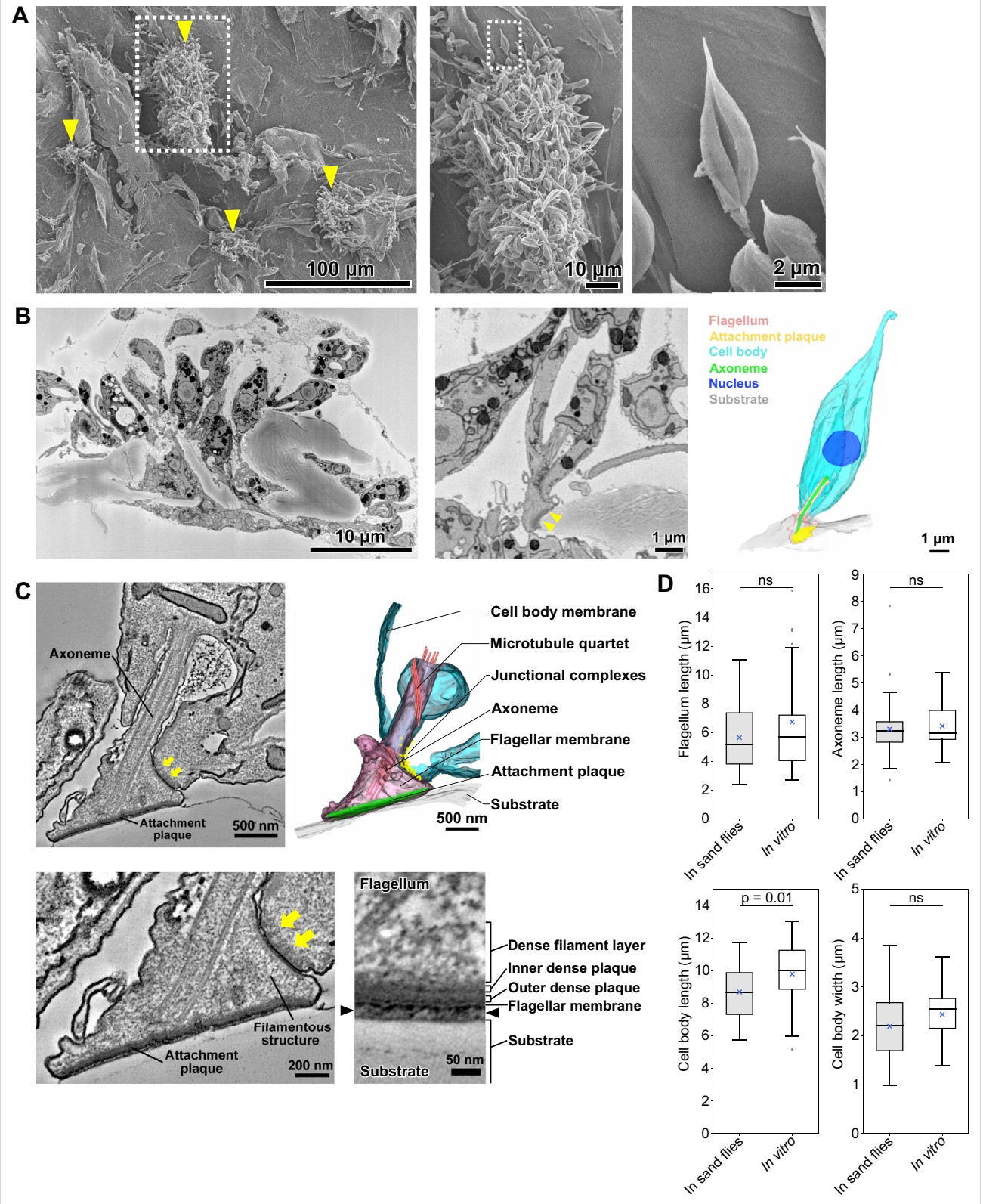

**Figure 3.** In vitro haptomonad-like cells resemble haptomonads observed in the sand fly. (**A**) Scanning electron microscopy images of in vitro haptomonad-like cells on a plastic coverslip. Left: Low-magnification image showing groups of cells (yellow arrowheads) attached to the substrate. Middle: a magnified view of the clump of attached haptomonad-like cells indicated by the white dotted box in the left image. Right: an individual haptomonad-like cell from the attached clump indicated by the white dotted box in the middle image. (**B**) Low-magnification SBFSEM image of

*Figure 3 continued on next page*

*Figure 3 continued*

haptomonad-like cells attached to a plastic coverslip in vitro (left; *Figure 3—video 1*). Magnified SBFSEM image (middle) and 3D reconstruction (right) of an in vitro haptomonad-like cell. SBFSEM slices show the haptomonad-like cell attached to the substrate with an electron-dense attachment plaque (yellow arrowheads). (**C**) Slices of a serial tomogram and 3D reconstruction (top right) of an in vitro haptomonad-like cell. Top left: tomographic slice through the region of the flagellum attachment to the cell, highlighting the FAZ and the flagellar pocket. Yellow arrows: junctional complexes. Bottom left: high-magnification view of the attachment region and the FAZ. Bottom right: magnified views of the tomogram showing that the attachment plaque observed in vitro has the same distinct ultrastructural elements identified in attachment plaques in vivo (in the sand fly; *Figure 2G*). Black arrowheads show the gap between the flagellum and the substrate. (**D**) Box-whisker plots of flagellum (top left), axoneme (top right) and cell body (bottom left) length and cell body width (bottom right) of haptomonads in sand flies and in vitro haptomonad-like cells. Boxes and whiskers indicate the median, upper and lower quartiles and 5th/95th percentiles. Blue crosses and grey dots indicate mean values and outliers, respectively. No significant (ns) morphological differences were found between haptomonads in sand flies and in vitro haptomonad-like cells, except for cell body length. p-Values calculated using two-tailed Welch's *t*-test. N=50 (in sand flies) and 30 (in vitro).

The online version of this article includes the following video and figure supplement(s) for figure 3:

**Figure supplement 1.** Cell cycle analysis of in vitro haptomonad-like cells.

**Figure supplement 2.** Sand fly haptomonads and in vitro haptomonad-like cells are surrounded by an extracellular filamentous gel-like matrix.

**Figure 3—video 1.** SBFSEM imaging of in vitro haptomonad-like cells attached to plastic.

https://elifesciences.org/articles/84552/figures#fig3video1

**Figure 3—video 2.** 3D model of mapping of nuclei of in vitro haptomonad-like cells.

https://elifesciences.org/articles/84552/figures#fig3video2

**Figure 3—video 3.** Serial section electron tomography imaging of an in vitro haptomonad-like cell attached to plastic.

https://elifesciences.org/articles/84552/figures#fig3video3

**Figure 3—video 4.** 3D model generated from the serial tomogram shown in *Figure 3—video 3*.

https://elifesciences.org/articles/84552/figures#fig3video4

cell tip, with an expanded FAZ area containing typical junctional complexes (*Figure 3—videos 3–4*, *Figure 3C*). Filamentous structures were present across the flagellum extending from the FAZ to the attachment plaque complex. The attachment plaque complex had a layered structure similar to that of the haptomonads in the sand fly, although the dense filament layer that is proximal to the dense plaque (towards the cell body) appeared slightly sparser in the in vitro haptomonad-like cell (*Figure 3C*).

In addition, we compared the morphology of in vitro haptomonad-like cells with the sand fly haptomonads by measuring flagellum and axoneme length, and cell body length and width using the SBFSEM data (*Figure 3D*). The in vitro haptomonad-like cells had a slightly longer cell body, but were similar in terms of flagellum and axoneme length and cell body width. Overall, the in vitro haptomonad-like cells appeared similar in cellular organisation and ultrastructure to the sand fly haptomonads, confirming that the in vitro forms are a good foundation on which to study the molecular cell biology of haptomonads.

## Adhesion of the in vitro haptomonad-like cell occurs through a series of defined steps

The plastic surface used for generation of in vitro haptomonad-like cell was suitable for electron microscopy analyses, but was not suitable for light microscopy. However, we discovered that specific glass substrates whose surfaces have hydrophobic properties supported the differentiation of promastigotes to in vitro haptomonad-like cells. We took advantage of this and followed the in vitro adhesion process using time-lapse microscopy (*Figure 4—video 1*; *Figure 4A*). The adhesion process in our system took between 2 and 9 hr to complete and did not appear synchronous in the population. After examining 8 videos of cells adhering (showing a total of 10 adhering cells), we identified five distinct stages in the adhesion process (*Figure 4—video 1*; *Figure 4A*). In the first stage of adhesion (stage 1), the cells appeared to explore the surface with their flagellum, making contact as shown by the distortion of the flagellar membrane and the release of membrane 'streamers' (*Figure 4—videos 1–2*; *Figure 4B*; *Figure 4—figure supplement 1*; *Ellis et al., 1976*; *Bloodgood, 1990*). In stage 2, a segment of the flagellum initiated adhesion to the substrate. The initial point of adhesion was not restricted to the tip of the flagellum, but could be any point along the flagellum length. Stage 3 was characterised by cells remaining more stably fixed to a specific point on the surface, with the cell being

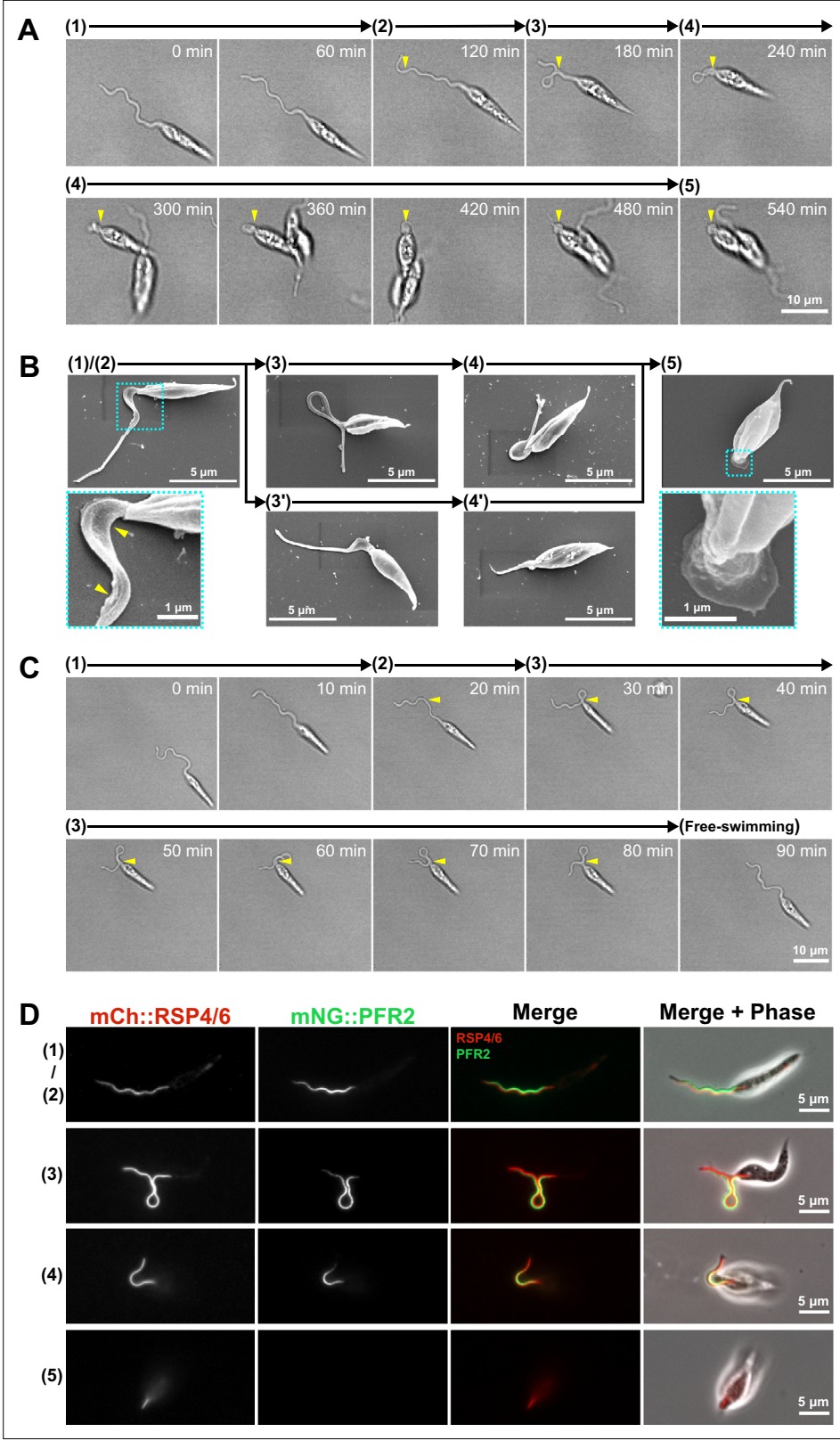

**Figure 4.** Adhesion of the in vitro haptomonad-like cell has a series of defined sequential steps. (**A**) Sequential frames (at 60 min intervals) from a time-lapse video (*Figure 4—video 1*) of adhesion of a haptomonad-like cell, as viewed by bright field microscopy. The numbers and arrows above the images indicate the 5 different stages of adhesion observed by time-lapse. Yellow arrowheads show the adhesion point. In stage 1, a promastigote cell

*Figure 4 continued on next page*

*Figure 4 continued*

first probes the substrate surface using the flagellum, and then a small section of the flagellum starts to adhere to the surface (stage 2). In stage 3, the flagellum is more clearly (and widely) adhered to the surface, and then the flagellum disassembles (stage 4). Finally, a clear attachment plaque is observed, representing mature attachment (stage 5). (**B**) Scanning electron microscopy images of cells in different stages of adhesion, showing that initial adhesion and disassembly of the flagellum can occur with (3 and 4) or without (3′ and 4′) the formation of a loop in the flagellum. The dotted boxes show magnified views of the attachment region in cells at early (1 and 2) and late (5) stages of adhesion. The yellow arrowheads indicate deformation of the flagellar membrane in initial adhesion regions. (**C**) Sequential frames (at 10 min intervals) from a time-lapse video (*Figure 4—video 3*) showing that adhesion is reversible at its early stages (stage 1–3). (**D**) Fluorescent images of promastigotes and haptomonad-like cells expressing an axoneme marker (mCherry::RSP4/6; red) and a PFR marker (mNeonGreen::PFR2; green) showing the disassembly of the PFR and partial disassembly of the axoneme. The stages of adhesion are indicated on the left-hand side of each image row.

The online version of this article includes the following video and figure supplement(s) for figure 4:

**Figure supplement 1.** *Leishmania* explores the surface while releasing membrane streamers from the flagellum.

**Figure supplement 2.** Measurement of mNeonGreen::PFR2 and mCherry::RSP4/6 length in in vitro haptomonad-like cells.

**Figure supplement 3.** In vitro purified metacyclics do not adhere to glass.

**Figure 4—video 1.** Time-lapse video of adhesion process of an in vitro haptomonad-like cell on a glass coverslip.
https://elifesciences.org/articles/84552/figures#fig4video1

**Figure 4—video 2.** Time-lapse video of release of membrane streamers from the flagellum of an in vitro haptomonad-like cell on a glass coverslip.
https://elifesciences.org/articles/84552/figures#fig4video2

**Figure 4—video 3.** Time-lapse video of reversible adhesion of an in vitro haptomonad-like cell on a glass coverslip.
https://elifesciences.org/articles/84552/figures#fig4video3

able to move relative to this point, while the attachment region appeared to slide along the length of the flagellum. In cells where the initial point of adhesion was away from the base of the flagellum, a flagellum loop often formed, ensuring that the cell body was in close proximity to the point of adhesion (in the videos examined, 7 out of 10 cells attempting to attach formed a flagellum loop). In stage 4, the flagellum began to shorten, and expansion of the flagellar membrane was observed, with fusion of the membrane in the looped area of the flagellum if present. Finally, in the last stage of adhesion (stage 5), there was maturation of the attachment region, with the presence of a clear attachment plaque and the cell body rotating from lying parallel to the surface to being upright. Throughout these steps, the cell body gradually became shorter and wider.

When examining cells that had been allowed to adhere to glass for 24 hr, we identified each of these stages described above by SEM (*Figure 4B*). This suggests that the adhesion process of in vitro haptomonad-like cells occurs through a defined series of steps. In addition, we noted that some attempts to attach to the surface were unsuccessful. In the videos examined, 3 out of 10 cells observed attempting to attach to the surface formed an initial adhesion but then detached and swam away before flagellum disassembly (*Figure 4—video 3*; *Figure 4C*). This suggests that in vitro at least the initial stages (1-3) of adhesion are reversible.

Our electron microscopy showed that, despite being much reduced, the haptomonad axoneme still contained some of the canonical non-microtubule components such as the radial spokes, while the PFR was absent (*Figure 2E*). To examine this at the molecular level in our in vitro system, we differentiated a cell line expressing a radial spoke protein (RSP4/6) tagged with mCherry (mCh) and a PFR protein (PFR2) tagged with mNeonGreen (mNG; *Wang et al., 2021*), and examined them by fluorescence light microscopy (*Figure 4D*). In promastigote cells, RSP4/6 and PFR2 had their expected localisations along the flagellum. As adhesion of the in vitro haptomonad-like cells progressed, the RSP4/6 and PFR2 signals shortened in parallel with the flagellum shortening (*Figure 4—figure supplement 2*). Eventually, in the very short flagellum of mature in vitro haptomonad-like cells, only a short line of RSP4/6 signal was consistently seen, with the PFR2 signal not seen in 30% (9/30) of these cells (*Figure 4—figure supplement 2*), suggesting that the PFR is disassembled during in vitro haptomonad-like cell adhesion.

The cells with lateral attachment and unattached cells around the stomodeal valve (*Figure 1E–F*) had a cell morphology similar to that of metacyclics or leptomonads. To address whether metacyclics are able to attach, in vitro purified metacyclics were allowed to attach to glass for 24 hr (*Figure 4— figure supplement 3*). After 24 hr very few metacyclics had attached in comparison to promastigotes, suggesting that in vitro metacyclics have little adhesive capacity.

### Adehsion of in vitro haptomonad-like cells is dependent on Ca²⁺

The formation of hemidesmosomes and desmosomes in multicellular organisms is regulated by $Ca^{2+}$ (*Trinkaus-Randall and Gipson, 1984*). In contrast, earlier studies reported that trypanosomatid adhesion to surfaces in vitro was resistant to divalent cation depletion (*Hendry, 1987*; *Molyneux*

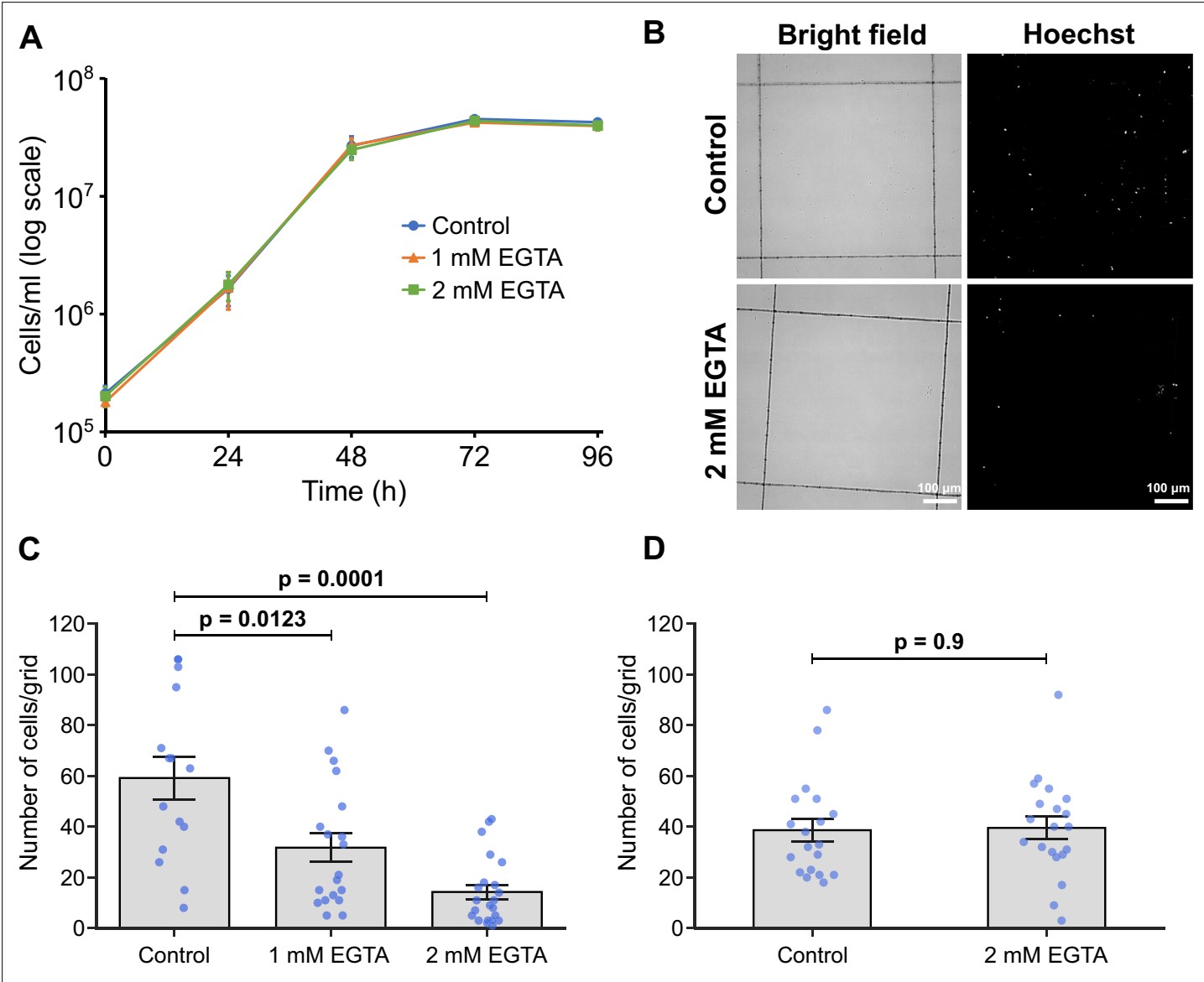

**Figure 5.** Calcium is necessary for adhesion of in vitro haptomonad-like cells. (**A**) Growth curve of cells cultured in control M199 growth medium or M199 with 1 or 2 mM EGTA. Data represent mean ± SD (n=3 independent experiments). (**B–D**) In vitro analysis of adhesion in the presence of EGTA. Cells were allowed to adhere to gridded coverslips for 24 hr, and subjected to EGTA treatment during (**B, C**) or after (**D**) adhesion. (**B**) Bright field and Hoechst fluorescence images of attached cells after 24 hr of culture in medium with or without 2 mM EGTA. (**C**) Quantification of the number of attached cells per grid area. Data represent mean ± SEM (n=3 independent experiments). (**D**) Quantification of the number of attached cells per grid area in attached cultures exposed to 2 mM EGTA (or control medium) for 30 min after 24 hr of adhesion. Data presented mean ± SEM (n=3 independent experiments). In C and D, the blue dots represent individual measurement from three-independent experiments. p-Values calculated using two-tailed Welch's *t*-test.

et al., 1987; Bloodgood, 1990). To investigate the role of Ca$^{2+}$ in the process of adhesion formation and maintenance, we examined adhesion in the presence of EGTA in M199 medium, which includes ~2 mM Ca$^{2+}$. The addition of 1 or 2 mM EGTA to the medium had no effect on the growth of promastigotes in suspension culture (Figure 5A). However, the presence of EGTA during adhesion resulted in a dramatic reduction in the number of attached cells after 24 hr (Figure 5B–C), with an increased effect at the higher concentration of EGTA (Figure 5C). These results contrast with those reported by Hendry, 1987, who found that T. congolense adheres to substrates in vitro even in Ca$^{2+}$/Mg$^{2+}$-free medium (medium containing 1 or 10 mM EDTA, or Ca$^{2+}$/Mg$^{2+}$-free PBS). Thus, our data indicate that there are different regulatory mechanisms for adhesion in different trypanosomatids. To examine whether the removal of Ca$^{2+}$ from cells that have already attached to the substrate disrupted their attachment, we incubated attached in vitro haptomonad-like cells (formed by differentiation for 24 hr in vitro) in the medium containing 2 mM EGTA for 30 min, and then counted the number of cells that remained attached to the glass (Figure 5D). The removal of Ca$^{2+}$ from the medium did not cause cells to detach (Figure 5D), which was consistent with the previous studies showing that Ca$^{2+}$ is not required for attachment maintenance in other trypanosomatids (Molyneux et al., 1987). Overall, our results suggest that Ca$^{2+}$ is critical for adhesion formation but not maintenance in Leishmania in vitro.

## Discussion

Strong attachment between cells and substrates is found across eukaryotic biology from the epithelium in mammals to unicellular eukaryotic parasites, including the kinetoplastid parasites. The ability to adhere and maintain position in a specific environment is important for the establishment and maintenance of an infection in the host and vector. The haptomonad form is an enigmatic stage in the life cycle of Leishmania, as its specific role is unclear and the adhesion mechanism has not been described in detail. Here, we have used a combination of volume electron microscopy approaches to define the 3D organisation and cytoskeletal architecture of haptomonad attachment connecting the parasite to the stomodeal valve. We have also analysed an in vitro system of adhesion, which has allowed us to define a discrete series of events leading to mature attachment.

Using SBFSEM, we examined the organisation of hundreds of haptomonad forms attached to a sand fly stomodeal valve. The parasites were densely packed on the stomodeal valve with several layers of haptomonads. As would be expected, those parasites in close proximity to the stomodeal valve had a shorter flagellum in comparison with those positioned further away from the surface of the valve. Haptomonads that cannot bring their cell bodies close to the stomodeal valve due to high cell density may position their cell bodies in the vacant space by adjusting their flagellum length. Alternatively, the positioning of the cells relative to the stomodeal valve may represent different stages along the haptomonad differentiation process, with those closest to the surface fully differentiated. This notion is in line with our time-lapse observations of the differentiation in which we saw a progressive disassembly of the flagellum, resulting in an in vitro haptomonad-like cell attached to the glass surface by a short, enlarged flagellum. For the majority of haptomonads examined the attachment interface was positioned at the enlarged distal tip of the flagellum; however, in a significant minority of cells the interface occurred along the side of the flagellum not the tip. Interestingly, the initial adhesion between the parasite and the glass surface occurred through the lateral face of the flagellum. These commonalities between the haptomonads in the sand fly and in vitro haptomonad-like cells further confirm the latter's suitability for future studies of haptomonad biology. However, it should be noted that as there are no molecular markers defining a haptomonad, we were not able to confirm on the molecular level whether our in vitro haptomonad-like cells are comparable to haptomonads in the sand fly.

The SBFSEM observations emphasise the high-volume spatial organisation and context of the haptomonads on the stomodeal valve. To add further detailed ultrastructural information on the attachment structure, we used electron tomography. This highlighted important changes to the cellular architecture of the haptomonad in comparison to the promastigote. There was a reduction in the length of the flagellar pocket neck in the haptomonad and a widening of the anterior cell tip, with an expansion of the flagellum as it exits the neck. It therefore appears as if the flagellar pocket neck has been 'unpeeled' and spread out to create a larger interface between the cell body and flagellar membranes mediated by the junctional complexes of the FAZ.

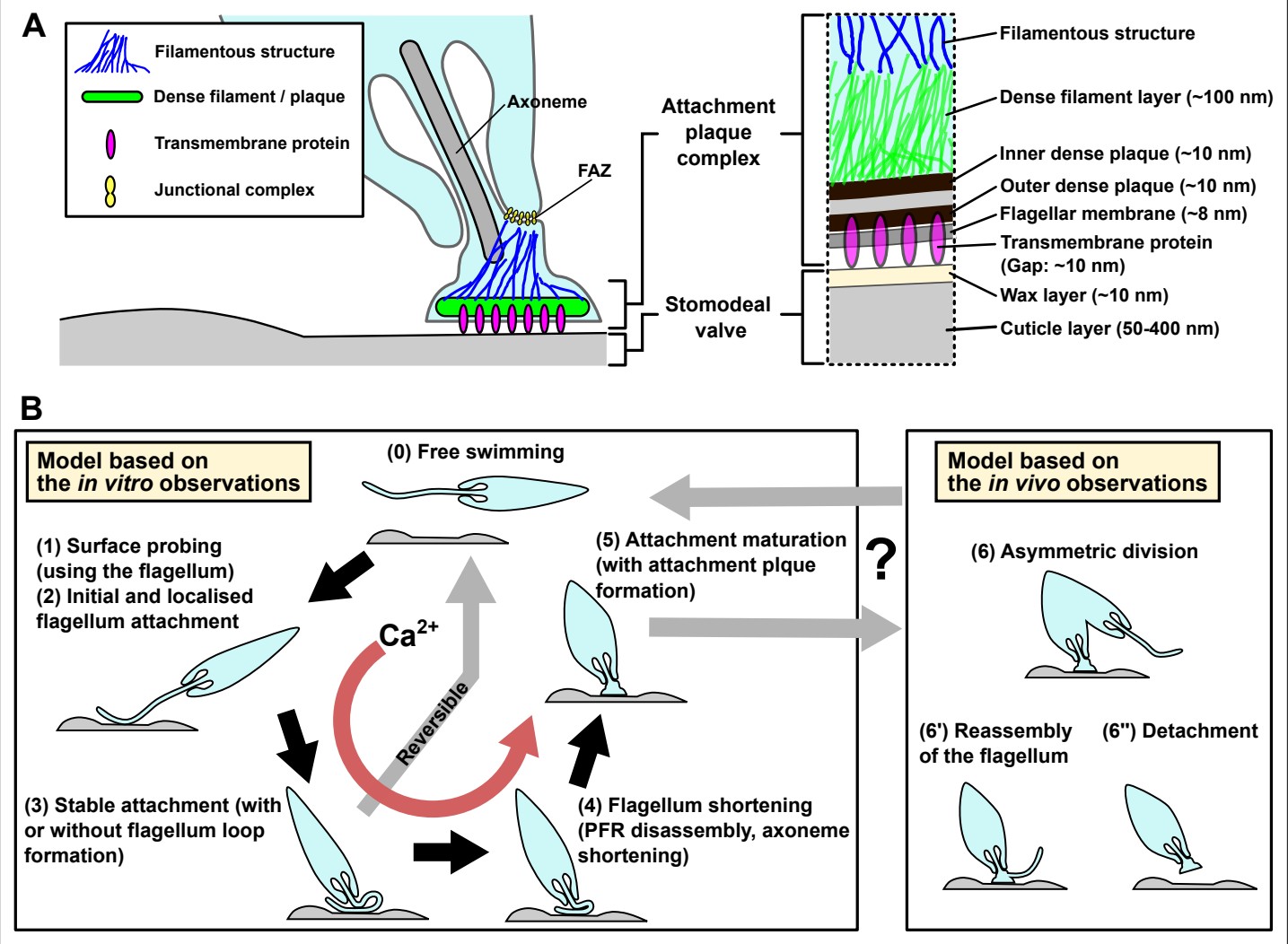

**Figure 6.** Models of the haptomonad attachment plaque and adhesion process. (**A**) Model of the haptomonad attachment plaque. (**B**) Model of the haptomonad adhesion process based on the in vitro observations. A promastigote cell explores the surface of the substrate using the flagellum (1), and initiates adhesion through the flagellum (2). Subsequently, stable adhesion is established (3), which is often (but not always) associated with the formation of a flagellum loop. Then, the flagellum becomes shorter (partial flagellum disassembly; 4), and finally the adhesion matures, with the formation of an attachment plaque (5). Stage 3 is reversible, as cells can still detach from the substrate before flagellum shortening. Adhesion formation (but not maintenance) is Ca²⁺-dependent. The box on the right in B shows three different hypotheses for generation of free-swimming promastigotes from haptomonads based on the in vivo observations, including asymmetric division (6), flagellum reassembly followed by detachment (6'), or detachment followed by flagellum reassembly (6'').

The haptomonad flagellum has a shortened axoneme that only just extends beyond the end of the cell body. However, unlike the amastigote flagellum, which has also disassembled the central pair microtubules, the haptomonad flagellum has retained its central pair and other accessory structures associated with motility such as the outer and inner dynein arms. Moreover, the presence of a radial spoke protein, RSP4/6, in the in vitro haptomonad-like flagellum was confirmed by protein tagging, suggesting that the in vitro haptomonad-like flagellum is still capable of movement, and indeed we observed movement of the shortened attached flagellum of in vitro haptomonad-like cells. Such motility if it occurs in the sand fly may contribute to the uptake of material into the flagellar pocket or to the damage of the stomodeal valve that occurs during *Leishmania* infection, or potentially the rapid assembly of a long motile flagellum on differentiation into promastigotes.

The electron tomography observations also revealed that the structural architecture of the haptomonad attachment plaque consisted of discrete layers, with the inner and outer dense plaques overlaying the flagellar membrane connecting to a dense filament layer, summarised in *Figure 6A*.

Interestingly, this is a similar arrangement to that of the hemidesmosome formed in mammalian epithelial cells when they attach to the basement membrane (*Todorović et al., 2004*). This has been noted before and previously the attachment plaque has been termed a hemidesmosome *Killick-Kendrick et al., 1974*; however, as no orthologues of the constituent proteins of the hemidesmosome are found in the *L. mexicana* genome, we use the term attachment plaque.

The thickness of the filaments in the haptomonad flagellum is 8–10 nm, which is similar to the thickness of intermediate filaments or septin filaments in other eukaryotes (*Bridges et al., 2014*; *Coulombe and Wong, 2004*). In addition, the thickness of the filament bundle is 16–20 nm and just as intermediate or septin filaments can form a bundle, the filament bundle in the haptomonad flagellum may be formed by bundling of these individual filaments. Negative staining of cytoskeletons of in vitro haptomonad-like cells also showed filaments bundled together near the FAZ and the attachment region (*Figure 2—figure supplement 1B*). However, again, no orthologues of intermediate filament and septin filament proteins, such as keratins or septins, are present in the *Leishmania* genome. *Leishmania* may therefore have unique filament-forming proteins constituting the filamentous structures in the haptomonad flagellum. Connections were observed in the gap between the surface of the stomodeal valve and the flagellar membrane. These connections spanned gaps with a constant spacing of ~10 nm, and were not seen in the regions where the gap was wider. This suggests that they have a specific size and are therefore more likely to represent transmembrane protein(s), rather than flexible substances such as mucus material. Overall, there has been a dramatic remodelling of the flagellum in the haptomonad. The strong attachment of the cell to the cuticle of the valve is not just mediated by the attachment plaque but is buttressed by the numerous filaments and filament bundles, which connect the attachment interface to the junctional complexes of the FAZ.

Numerous vesicles and vesicular structures were seen within the flagellum. The disassembly of the flagellum during haptomonad differentiation requires a reduction in the amount of flagellar membrane and these vesicles may represent the mechanism by which this membrane is removed and transported back to the cell body, albeit recognising the potential impediment caused by basal body and associated structures. However, in addition to these larger vesicular structures there were smaller vesicles seen in close proximity to the attachment interface, with one in the process of fusing to or budding from the flagellar membrane at this point. The attachment plaque is a complex structure associated with a specialised membrane domain requiring many different components that need to be delivered to the attachment interface for assembly. It is entirely feasible that these components are packaged into vesicles before delivery to the interface and not without precedence as the assembly of desmosome connections between mammalian cells occurs through the delivery of vesicles loaded with desmosome proteins in a calcium dependent manner (*Burdett and Sullivan, 2002*). Alternatively, while the flagellar pocket in *Leishmania* is generally thought of as the only site of exo/endocytosis, the attachment interface could potentially support these processes and may be important for the release of factors such as chitinase directly onto the cuticle surface of the stomodeal valve.

Previous work had suggested that the *L. mexicana* haptomonad form was non-replicative (*Gossage et al., 2003*) and while the majority of haptomonads we examined only had one flagellum and one nucleus, we saw examples of cells dividing. The fate of the daughter cells from these divisions has yet to be determined; however, they could represent an asymmetric differentiation division as seen in other kinetoplastids in which one of the daughter cells differentiates to become a different life cycle stage (*Peacock et al., 2018*; *Rotureau and Van Den Abbeele, 2013*; *Skalický et al., 2017*). Yet, in a dividing cell with a long new flagellum both the old and new flagella were attached to the surface, suggesting that this is a proliferative division, with both daughter cells remaining attached. In addition, in those dividing cells with a short new flagellum there was no evidence of any attachment structures, suggesting that this is post-axonemal assembly modification and requires the flagellum to assemble to a length able to engage with the stomodeal valve before this process begins. Intriguingly, we also observed cells that contained two or three nuclei but in which flagellar division was not occurring and these may represent cells from fusion events, malformed divisions, or steps in gametogenesis. Sexual recombination in *Leishmania* has been shown to occur in the sand fly and the close positioning of attached haptomonads may facilitate the requisite cell fusion events (*Akopyants et al., 2009*; *Inbar et al., 2013*; *Peacock et al., 2011*; *Sadlova et al., 2011*; *Serafim et al., 2022*).

We have devised a model to summarise our current understanding of the process of *Leishmania* adhesion, by combining these observations from our volume electron microscopy from the sand fly

derived haptomonads and light microscopy of the in vitro haptomonad-like cells, with the caveat that the in vitro processes may not perfectly replicate the sand fly situation (*Figure 6B*). A free-swimming promastigote cell first starts to explore the surface of the substrate using its flagellum. If a suitable position is found the adhesion process is triggered — what this signal is remains unknown, but calcium potentially plays a role, as the removal of calcium inhibited cell adhesion. The cell then establishes the initial adhesion, which was often associated with the formation of a loop of flagellum. Currently, we do not know whether the promastigote to haptomonad differentiation is reversible or not. In our time-lapse videos, we saw promastigotes that would adhere to the surface but would not fully adhere and instead detach and swim off. This suggests that the initial stages of adhesion in vitro at least are reversible. Around this stage, however, a point of no return occurs and the parasites from our in vitro data were unable to detach from the surface. Adhesion continues with the disassembly of the axoneme and PFR, expansion of the flagellar membrane and the assembly of the attachment plaque with the cell body reducing in length and widening. The long axis of the mature in vitro haptomonad-like cell then becomes orthogonal to the surface, with the cell standing proud attached through its flagellum tip. From our SBFSEM data, the haptomonad form is then able to divide at a very slow rate, generating more attached forms and potentially unattached forms through an asymmetric division. Moreover, we saw examples of cells that had become detached from the stomodeal valve, with a clear attachment plaque density at their enlarged flagellum tip that was not in contact with the stomodeal valve surface. This may represent the first step of the differentiation process of haptomonads into free-swimming forms. Many cells with lateral flagellum attachment to the stomodeal valve and unattached cells were morphologically similar to leptomonads or metacyclics. Our analysis of in vitro purified metacyclics suggest they are not competent to adhere. If the metacyclics in the sand fly were also not adhesion competent then those cells attached to the valve laterally through their flagellum may represent leptomonad cells in the process of differentiating to the haptomonad form as suggested in the previous study (*Rogers et al., 2002*) or haptomonads differentiating to metacyclics or leptomonads.

The detailed 3D organisation and architecture of the haptomonad attachment and its process of adhesion development revealed in this study will be of great help for future work on the identification of proteins involved in adhesion and understanding the mechanism of the haptomonad attachment and its role in the *Leishmania* life cycle.

# Materials and methods

**Key resources table**

| Reagent type (species) or resource | Designation | Source or reference | Identifiers | Additional information |
|---|---|---|---|---|
| Gene (*Leishmania mexicana*) | RSP4/6 | TriTrypDB (http://tritrypdb.org/tritrypdb/) | LmxM.13.0430 | |
| Gene (*Leishmania mexicana*) | PFR2 | TriTrypDB (http://tritrypdb.org/tritrypdb/) | LmxM.16.1430 | |
| Strain, strain background (*Leishmania mexicana*) | *L. mexicana* | Sunter lab stocks | WHO strain MNYC/BZ/1962 /M379 | The identity has been authenticated by genome and mRNA sequencing. The strain was monitored for contamination, including mycoplasma contamination, through DNA staining and microscopy during data capture. |
| Cell line (*L. mexicana*) | mCherry::RSP4/6, mNeonGreen::PFR2 | *Wang et al., 2021* *L. mexicana* cell line provided by Dr Richard Wheeler (University of Oxford) | NA | The identity has been authenticated by genome sequencing. The cell line was monitored for contamination, including mycoplasma contamination, through DNA staining and microscopy during data capture. |
| Strain, strain background (*Lutzomyia longipalpis*) | *L. longipalpis* | *Volf and Volfova, 2011* | NA | |
| Software, algorithm | 3dmod | *Kremer et al., 1996* | PMID:8742726 | |

*Continued on next page*

*Continued*

| Reagent type (species) or resource | Designation | Source or reference | Identifiers | Additional information |
|---|---|---|---|---|
| Software, algorithm | IMOD | http://bio3d.colorado.edu/imod | RRID:SCR_003297 | |
| Software, algorithm | SerialEM | *Mastronarde, 2005* | PMID:16182563 | |
| Software, algorithm | Fiji | *Schindelin et al., 2012* | RRID: SCR_002285 | |
| Software, algorithm | Excel | https://microsoft.com/mac/excel | RRID:SCR_016137 | |
| Software, algorithm | python | https://www.python.org/ | RRID:SCR_008394 | |
| Software, algorithm | matplotlib | *Hunter, 2007* | RRID:SCR_008624 | |
| Other | Gridded glass coverslips grid-500 | iBidi | Cat. #: 10816 | In vitro haptomonad-like cell preparation |
| Other | Thermanox plastic coverslips | Nalgene Nunc International | Cat. #: 174950 | In vitro haptomonad-like cell preparation |
| Other | μ-dish 35 mm, high grid-500 glass bottom | iBidi | Cat. #: 81168 | In vitro haptomonad-like cell preparation |

## Cell culture

*L. mexicana* (WHO strain MNYC/BZ/1962 /M379) promastigotes were grown at 28 °C in M199 medium with 10% foetal calf serum (FCS), 40 mM HEPES-HCl (pH 7.4), 26 mM NaHCO$_3$ and 5 µg/ml haemin. Cells were maintained in logarithmic growth by regular subculturing.

## Infection of *L. mexicana* in the sand fly

Sand fly infection was carried out as described in *Sádlová et al., 2021*. Briefly, females of *Lutzomyia longipalpis* were fed through a chick-skin membrane on heat-inactivated sheep blood containing *Leishmania mexicana* promastigotes from log-phase cultures at a concentration $2\times10^6$ cells/ml. Blood-engorged females were separated and maintained at 26 °C and high humidity with free access to a 50% sugar solution, with a 14 hr light/10 hr dark photoperiod. They were dissected on day 10 after a bloodmeal, and the dissected guts were fixed for 24 hr at 4 °C in Karnovsky fixative (2.5% glutaraldehyde and 2% paraformaldehyde in 0.1 M cacodylate buffer (pH 6.9)), transferred to the washing solution (0.1 M cacodylate buffer with 2.7% glucose) and kept at 4 °C.

## Serial block face scanning electron microscopy (SBFSEM)

All steps of sample processing for SBFSEM were performed at room temperature, unless stated otherwise, and all washing steps consisted of 3 washes, with 5 min incubations/wash. The fixed *Leishmania* infected guts were washed in 0.1 M cacodylate buffer (pH 6.9) and then incubated in 1% osmium tetroxide in 0.1 M cacodylate buffer containing 1.5% potassium ferricyanide, for 1 hr in the dark. The guts were then washed with ddH$_2$O, incubated in freshly prepared 1% thiocarbohydrazide (TCH) for 20 min in the dark, washed in ddH$_2$O, and incubated in 2% osmium tetroxide in ddH$_2$O for 30 min in the dark. The guts were washed again with ddH$_2$O and incubated in 1% uranyl acetate in ddH$_2$O overnight, at 4 °C and in the dark. Then, the guts were washed with ddH$_2$O and dehydrated in an ethanol series (30, 50, 70, 90, 100% (v/v), and 2×absolute ethanol; 10 min / step). The guts were embedded in TAAB 812 hard resin (TAAB Laboratories Equipment Ltd, Aldermaston, UK). Resin pieces containing the stomodeal valve region of the gut were mounted onto aluminium pins using conductive epoxy resin, sputter coated with a thin (12–14 nm) layer of gold, and then imaged in a Zeiss Merlin VP Compact fitted with a 3view2XP system (Gatan/Ametek, Pleasanton, CA). Serial images of the block face were recorded at an accelerating voltage of 1.8 kV and an aperture size of 20 µm. The pixel size and the dwell time for image capture were 5 nm and 3 µs, respectively, and the slice thickness was 100 nm. Images were recorded using an OnPoint backscattered electron detector (Gatan/Ametek, Pleasanton, CA). Data were segmented manually to 3D models, using 3dmod (IMOD software package; *Kremer et al., 1996*; see below for more detailed method). For the SBFSEM observation of in vitro haptomonad-like cells, plastic coverslips containing adherent cells were fixed in 0.1 M

phosphate buffer (pH 6.9) with 2.5% glutaraldehyde and 2% paraformaldehyde for 2 hr, and then embedded as described above. After resin hardening, the plastic coverslip was removed and samples were remounted so that the attached surface was surrounded by resin on both sides. Resin pieces containing the attachment region were mounted onto pins with the attachment surface perpendicular to the block face, and then imaged as described above.

## Serial section electron microscopy tomography

Ribbons of serial sections of ~150 nm were produced from sample blocks prepared for SBFSEM observations (as described above), and collected on formvar-coated slot grids. Sections were stained with Reynolds lead citrate before imaging at 120 kV, on a Jeol JEM-1400Flash (JEOL, Akishima, Japan) with a OneView (Gatan/Ametek, Pleasanton, CA) camera. Each individual tomogram was produced from a total of 240 4K x 4K pixel images (120 tilted images each of 0 and 90° axes, with 1° tilting between images) acquired automatically using SerialEM (*Mastronarde, 2005*; *Mastronarde, 2003*). Individual tomograms were produced using ETomo (IMOD software package), and consecutive tomograms were then joined to produce serial tomogram volumes, using ETomo. Data were segmented manually to produce 3D models, using 3dmod (see below for more detailed method).

## Three-dimensional modelling and data deposition of the SBFSEM and electron tomography data

Three-dimensional models of the SBFSEM and electron tomography data were created by manually tracing structures in each slice, and surface meshes were computed from the traced contours in each slice using 3dmod (see the following link for more detailed manual; https://bio3d.colorado.edu/imod/doc/3dmodguide.html). The 3D models of the stomodeal valve or plastic substrate with high electron density outlines were generated using the isosurface function of 3dmod. Also, we deposited the original SBFSEM (EMPIAR-11463 and EMPIAR-11464) and electron tomography (EMPIAR-11467 and EMPIAR-11468) data as an MRC image file, which can be opened and analysed using 3dmod, on EMPIAR (*Iudin et al., 2023*; https://www.ebi.ac.uk/empiar/).

## In vitro haptomonad-like cell adhesion

Axenic haptomonad-like cells were generated by culturing $1 \times 10^6$ cells/ml promastigotes on gridded glass coverslips grid-500 (iBidi, Gräfelfing, Germany) which were cut into small pieces of ~5 × 5 mm and sterilised with 100% ethanol (for light microscopy and SEM) or 13 mm round Thermanox plastic coverslips (Nalgene Nunc International, Rochester, NY) scratched with sandpaper and sterilised with 100% ethanol (for SBFSEM) in a 24-well plate with 1 ml of M199 medium at 28 °C with 5% $CO_2$ for 24 hr (for light microscopy and SEM) or 72 hr with M199 medium being replaced every 24 hr (for SBFSEM).

## Light microscopy

For light microscopy of living cells, in vitro haptomonad-like cells attached to a piece of a gridded glass coverslip were washed twice in DMEM, incubated in DMEM with Hoechst 33342 (1 µg/ml) for 5 min and then washed twice in DMEM. Coverslip pieces were mounted onto another glass coverslip and then onto a glass slide, with the cell attachment side facing up. Attached cells were imaged using a Zeiss ImagerZ2 microscope with 63×objective and a Hamamatsu Flash 4 camera.

## Time-lapse observation of in vitro haptomonad-like cell adhesion

For time-lapse observation, log phase promastigotes ($1 \times 10^6$ cells/ml) were cultured in a µ-dish 35 mm, high grid-500 glass bottom (iBidi, Gräfelfing, Germany) for 12 hr, and the dish was washed five times with fresh M199 medium before the start of imaging. Cells about to adhere to the glass were recorded using Zeiss LSM 880 confocal microscopy with 63×objective for 24 hr at 28 °C with 5% $CO_2$ in a chamber with controlled temperature and $CO_2$ concentration.

## Scanning electron microscopy

In vitro haptomonad-like cells on a piece of a gridded glass coverslip were fixed with 2.5% glutaraldehyde in PEME (0.1 M PIPES, pH 6.9, 2 mM EGTA, 1 mM $MgSO_4$, 0.1 mM EDTA). After an hour, coverslips were washed once in PEME and once in $ddH_2O$. The coverslips were then dehydrated

using increasing concentrations of ethanol (30%, 50%, 70%, 90%, 100% (v/v), and 2×absolute ethanol; 10 min / step). The coverslips were then critical point dried, mounted onto SEM stubs using carbon stickers, and sputter coated with a layer of 12–14 nm of gold. Images were taken on a Hitachi S-3400N scanning electron microscope at 5 kV, at a 5.5 mm working distance.

## Quantitative analysis of the effect of calcium on in vitro haptomonad-like adhesion

Log phase promastigotes ($1\times10^6$ cells/ml) were cultured on ~5 × 5 mm pieces of gridded glass coverslips in a 24-well plate with 1 ml of control M199 medium (complete M199 medium including 10% ddH$_2$O) or M199 media including 10% ddH$_2$O and 1 or 2 mM EGTA (all media were adjusted to pH 7.4 with NaOH) for 24 hr at 28 °C with 5% CO$_2$. The coverslips were washed twice with 1 ml of DMEM, incubated in 1 ml of DMEM with Hoechst 33342 (1 µg/ml) for 5 min, and washed twice with 1 ml of DMEM. The glass pieces were mounted with another glass coverslip on a glass slide. The cells were imaged using a Zeiss ImagerZ2 microscope with 20×objective and Hamamatsu Flash 4 camera.

Log phase promastigotes ($1\times10^6$ cells/ml) were cultured on ~5 × 5 mm pieces of gridded glass coverslips in a 24 well plate with 1 ml of M199 medium for 24 hr at 28 °C (in a 5% CO$_2$ atmosphere). The coverslips were transferred into 1 ml of control M199 medium or M199 medium with 2 mM EGTA, respectively and incubated for 30 min. The coverslips were washed twice with 1 ml of DMEM, incubated in 1 ml of DMEM with Hoechst 33342 (1 µg/ml) for 5 min, and washed twice with 1 ml of DMEM. The glass pieces were mounted with another glass coverslip on a glass slide. The cells were imaged using a Zeiss ImagerZ2 microscope with 20×objective and Hamamatsu Flash 4 camera. To quantify cell adhesion under different conditions, the number of cells attached to a single grid lattice area was counted manually using Fiji (*Schindelin et al., 2012*).

## Negative staining of in vitro haptomonad-like cells

Promastigotes ($1\times10^6$ cells/ml) were cultured on formvar-coated 200 mesh nickel grids in a 6 well plate with 5 ml of M199 medium, for 24 hr at 28 °C (with 5% CO$_2$). In vitro haptomonad-like cells attached on the formvar membrane were treated with 1% IGEPAL in PEME (0.1 M PIPES, pH 6.9, 2 mM EGTA, 1 mM MgSO$_4$, 0.1 mM EGTA) for 5 min, fixed with 2.5% glutaraldehyde in PEME for 10 min and stained with 1% aurothioglucose in ddH$_2$O. The samples were observed using a Jeol JEM-1400 Flash transmission electron microscope operating at 120 kV and equipped with a OneView 16-megapixel camera (Gatan/Ametek, Pleasanton, CA).

## Metacyclic purification by Ficoll density gradient

In vitro L. mexicana metacyclics were purified as described in *Späth and Beverley, 2001*. Briefly, stationary phase parasites (~5 × 10$^7$ cells/ml) were centrifuged at 800 *g* for 10 min and the cells were resuspended in DMEM. In a 15 ml of Falcon tube containing 2 ml of 40% Ficoll 400 (Melford, Ipswich, Suffolk, UK) in ddH$_2$O at the bottom, 2 ml of 10% Ficoll 400 in M199 medium without FCS and then 2 ml of the concentrated parasites in DMEM were overlaid. The tube was centrifuged at 1300 *g* for 10 min and parasites located at the upper 10% Ficoll boundary were collected, centrifuged at 800 *g* for 5 min and resuspended in complete M199 at a cell concentration of $1\times10^6$ cells/ml. Log phase promastigotes and in vitro purified metacyclics ($1\times10^6$ cells/ml, respectively) were cultured on gridded glass coverslips in a 24 well plate with 1 ml of complete M199 medium for 24 hr at 28 °C with 5% CO$_2$, and the number of attached cells per grid area was quantified in the same method as described above.

## Statistical analysis

Means, SDs and SEMs were calculated using Microsoft Excel. Statistical significance was determined using two-tailed Welch's *t-test* carried out with Microsoft Excel. Differences were considered significant at the level of $p<0.05$. Data were plotted with Microsoft Excel or the Matplotlib package in Python (*Hunter, 2007*).

## Acknowledgements

We thank Dr Richard Wheeler (University of Oxford) for providing the *Leishmania mexicana* strain (mCherry::RSP4/6, mNeonGreen::PFR2) and the sand fly cartoon; the Oxford Brookes Centre for Bioimaging for the kind support in electron and light microscopy; Prof Keith Gull (University of Oxford), Prof Sue Vaughan (Oxford Brookes University) and members of the Sunter and Vaughan laboratory for discussions and comments on the manuscript. RY is supported by a JSPS Overseas Research Fellowship and NIBB Collaborative Research Program (20-515). This project has also received resources funded by the European Union's Horizon 2020 research and innovation programme under grant agreement No 731060 (Infravec2). Work in the lab of JDS is supported by the Wellcome Trust (221944/Z/20/Z). JS, BV, KP and PV were supported by ERD funds, project CeRaViP (16_019/0000759) and Czech Science Foundation (GACR 21–15700 S).

## Additional information

### Funding

| Funder | Grant reference number | Author |
| --- | --- | --- |
| Japan Society for the Promotion of Science | | Ryuji Yanase |
| National Institute for Basic Biology | 20-515 | Ryuji Yanase |
| Horizon 2020 Framework Programme | Infravec2 | Petr Volf |
| Wellcome Trust | 221944/Z20/Z | Jack D Sunter |
| European Regional Development Fund | CeRaViP | Petr Volf |
| Czech Science Foundation | GACR 21-15700S | Petr Volf |

The funders had no role in study design, data collection and interpretation, or the decision to submit the work for publication. For the purpose of Open Access, the authors have applied a CC BY public copyright license to any Author Accepted Manuscript version arising from this submission.

### Author contributions

Ryuji Yanase, Conceptualization, Data curation, Formal analysis, Funding acquisition, Validation, Investigation, Visualization, Methodology, Writing - original draft, Writing – review and editing; Flávia Moreira-Leite, Resources, Data curation, Formal analysis, Validation, Investigation, Visualization, Methodology, Project administration, Writing – review and editing; Edward Rea, Resources, Data curation, Validation, Investigation, Methodology, Project administration; Lauren Wilburn, Methodology; Jovana Sádlová, Resources, Funding acquisition, Validation, Investigation, Methodology; Barbora Vojtkova, Katerina Pružinová, Funding acquisition, Validation, Investigation, Methodology; Atsushi Taniguchi, Shigenori Nonaka, Resources, Formal analysis, Validation, Methodology, Writing – review and editing; Petr Volf, Resources, Supervision, Funding acquisition, Validation, Methodology, Project administration, Writing – review and editing; Jack D Sunter, Conceptualization, Resources, Formal analysis, Supervision, Funding acquisition, Methodology, Writing - original draft, Project administration, Writing – review and editing

### Author ORCIDs

Ryuji Yanase ⓘ http://orcid.org/0000-0001-9419-398X
Petr Volf ⓘ http://orcid.org/0000-0003-1790-1123
Jack D Sunter ⓘ http://orcid.org/0000-0002-2836-9622

### Decision letter and Author response

Decision letter https://doi.org/10.7554/eLife.84552.sa1
Author response https://doi.org/10.7554/eLife.84552.sa2

## Additional files

### Supplementary files
• MDAR checklist

### Data availability
All data generated or analysed during this study are included in the manuscript.

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
