## [Editor Report]

This important work substantially advances our understanding of the organization and architecture of haptomonads, a distinct and poorly understood developmental form of Leishmania found in sand fly vectors at later stages of infection. The 3D electron microscopy methodology, including serial block face scanning electron microscopy and electron tomography to visualize haptomonads in sand fly, is exceptional and establishes a new standard in the field.

---

## [Decision Letter]

**Decision letter after peer review:**

Thank you for submitting your article "Formation and three-dimensional architecture of Leishmania adhesion in the sand fly vector" for consideration by *eLife*. Your article has been reviewed by 2 peer reviewers, and the evaluation has been overseen by a Reviewing Editor and Anna Akhmanova as the Senior Editor. The reviewers have opted to remain anonymous.

Essential revisions:

1) Calcium dependency of the attachment to the valve in infected flies should be addressed. In addition, it will also be important to test its effects on transmission by bite. If the authors could provide evidence to support the claim that haptomonads are important to transmission, this would be a great addition to the field.

2) Apart from sharing structural elements of the attachment plaque, there is no evidence that the forms that attach in vitro represent an authentic and unique life cycle stage in vivo. If by "haptomonad", the authors are referring to any promastigote that attaches to a surface via the formation of a flagellar attachment plaque, then this should be explained and clarified in the manuscript.

*Reviewer #1 (Recommendations for the authors):*

While the studies lend impressive details to a previously described aspect of promastigote colonization of the stomodeal valve, they do not further our understanding of the consequences of this colonization, if any, to the ability of infected flies to transmit infection by bite. The degeneration of the valve tissue, including a detachment of the cuticle lining that was previously described in infected flies (Schlein et al., PNAS, 1992), was not observed in the current studies. As damage to the valve, mediated by a parasite-derived chitinase, was proposed to facilitate transmission by bite, some discussion as to why they fail to see any damage is warranted.

Is 'haptomonad' a term used to describe any promastigote seen in attachment to the stomodeal valve, regardless of morphology? The attached, yellow-colorized cell shown in Figures1B and F with an elongated cell body and extended axoneme has the appearance of a metacyclic form. Would not large numbers of metacyclic promastigotes be expected to colonize the anterior midgut in close proximity to the valve? This raises the question as to whether there is a dedicated stage that is pre-adapted to attachment. This could be addressed in part using the in vitro system. Will metacyclics purified from culture attach in vitro, and do they remodel their flagellum and assume a haptomonad morphology?

The calcium2+ dependency of the attachment was confined to the in vitro model. Can the divalent cations be chelated in vivo, perhaps by including EGTA in the sugar meals to address this important question in infected sandflies?

*Reviewer #2 (Recommendations for the authors):*

Line 29-30: haptomonad is indeed a possible critical form for transmission, but it lacks formal demonstration still in all literature available.

Lines 74-75: The literature cited here by the authors provides an overview of promastigotes attachment and flagellar modifications inside the sand fly vector. At Wakid and Bates, 2004, the different surfaces for in vitro attachment were proposed but no haptomonad was specifically implicated or status of differentiation was proposed and only "promastigotes" were termed. It does show the formation of hemidesmossomes in vitro but this is no guarantee that any promastigote can't do it or isn't an artifact from in vitro conditions. Due to that, authors cannot claim with the data provided that it's a protocol for having haptomonads in vitro. The manuscript should be adjusted to harbor accurate discussion and conclusions. How do the authors guarantee the hemidesmosomes found in vitro are comparable to haptomonads in vivo ones?

Line 87: Although haptomonads are predicted to have an important role in transmission, no data is available demonstrating its role. Too much to conclude here unless a formal demonstration is provided.

Line 138: how do the authors know the valve is heavily infected as no counts or infection status were provided?

Lines 145-147: Can this be due to sample prep/processing? The asymmetry?

Line 191: Vide comment for lines 74-75.

Lines 196-200: How the authors are sure of the structure being a clump of haptomonads? And not just an aggregate of cells in multiplication as normally found in cultures?

Line 227: The discussion on this topic should be carefully revised. For example, video 6 shows an in vitro promastigote attaching to the surface. This attachment has a hemidesmosome structure formation that resembles the in vivo haptomonad ones. However, this is no evidence that haptomonad differentiation is happening in vitro.

Lines 259-269: This section of the manuscript gives interesting information on the location of RSP4/6 and PFR2 on the attaching promastigotes to the scratched surface. Again, it is not conclusive of a haptomonad differentiation situation. The lack of PRF signal can only be concluded by the data presented as "disassembled during promastigote attachment and hemidesmosome formation". This is important information and I suggest the authors explore the strength of the data.

Lines 271-291: Same issue as before. To claim in vitro haptomonad formation the authors will need a more detailed comparison with fly-derived haptomonads. This is extremely challenging to achieve though. However, the lack of material does not allow for in vitro assumptions claimed in the manuscript. The current manuscript will be significantly more accurate if the authors focus only on the data and interesting details about the flagellum attachment to the surface that "can" be similar to the real scenario inside the sand fly.

The discussion that follows from here should be adjusted based on the comments above. The data is important and provide details not explored. Claiming it's a haptomonad process in vitro is only damaging the dataset as this cannot be concluded. Also, this manuscript will benefit a lot by having a detailed description of how to analyze and get to the 3D models presented. No detailed description was provided. This had a strong potential for usage beyond the Leishmania/sand fly field.

Line 478: Haptomonad is a cryptic life stage with an unknown ontology.

How can the authors guarantee the in vitro cells have a similar differentiation pattern to haptomonads found in vivo?

Line 487: Any media formulation changes will likely promote drastic changes on parasite growth and differentiation. Why for in vitro live cell imaging the authors changed the culture media conditions?

Why no statistics detailed methods part is available?

---

## [Author Response]

Essential revisions:1) Calcium dependency of the attachment to the valve in infected flies should be addressed. In addition, it will also be important to test its effects on transmission by bite. If the authors could provide evidence to support the claim that haptomonads are important to transmission, this would be a great addition to the field.

We agree with the reviewers that understanding the role of calcium for parasite attachment to the stomodeal valve in the fly is important. We anticipate that this would involve time-resolved local release from local calcium stores and not a global requirement for calcium; therefore, we always felt the global chelation of calcium would be uninformative even if it had an effect. in vitro we can test the effect of calcium removal in a defined system of cells in culture medium attaching to glass; however, the effect of EGTA within a complex biological system such as the gut of an infected sand fly would be very difficult to interpret. For example, the sugar meal does not go directly into midgut, it is stored in crop and released to midgut in small quantities 1-2 days post-feeding. We have no idea about stability of EGTA in sand flies and it would be impossible to control the quantity imbibed by sand flies, so the concentration and bioavailability within the midgut would be unknown and variable. Moreover, calcium chelation will likely have effects upon the insect itself that might influence parasite attachment and there are other *Leishmania* proteins such as GP63 that require calcium and are important for parasite development in the sand fly. However, despite these concerns, we did do this experiment, just in case there was an effect (Author response image 1–Author response image 3). The result was negative and given the many caveats we outlined this is unsurprising.

**Author response image 1. sa2fig1:** Infection rates and intensities of infections on days 6 and 9 post-infection (PI). Numbers above the bars indicate the number of dissected females. Levels of Leishmania infections were graded into four categories: negative, light (< 100 parasites/gut), moderate (100-1000 parasites/gut) and heavy (>1000 parasites/gut).

**Author response image 2. sa2fig2:** Localization of infections on days 6 and 9 post-infection (PI). Abdominal midgut (AMG), thoracic midgut (TMG) and stomodeal valve (SV). Numbers above the bars indicate the number of evaluated (positive) females.

**Author response image 3. sa2fig3:** L. mexicana haptomonads attached to the stomodeal valve (SV) Day 9 PI. A, B Light and fluorescence microscopy of infected sand flies fed with sugar. C, D Light and fluorescence microscopy of infected sand flies fed with sugar with 20 mM EGTA.

As a lab, we are focused on defining the role of the haptomonad within the parasite’s life cycle and its role in transmission from the sand fly to the host is of major interest to us. However, the approaches we developed in this manuscript are not appropriate to test the importance of haptomonads for onwards transmission in a meaningful manner, as we have been unable to disrupt attachment in the sand fly. Moreover, the focus in this paper is on the three-dimensional structural information of haptomonads and the details of the adhesion formation. Thus, we think that transmission experiments are much beyond the scope of this study, which is the first of its kind to address how attachment can possibly happen. Moreover, it is important to understand the nature of attachment before interfering with it in vivo in complex biological contexts and we have now defined the plaque in vivo and in vitro facilitating future studies.

Methods: Females of Lutzomyia longipalpis were fed through a chick-skin membrane on heat-inactivated sheep blood containing Leishmania mexicana (SMP1::mCherry) promastigotes from log-phase cultures (day 3-4) at a concentration 106 cells/ml. Blood-engorged females were separated and maintained at 26°C with free access to sugar solution only or sugar solutions containing 5 mM or 20 mM EGTA; all sugar solution contained blue food colour to be sure that dissected females took the sugar solution. On days 6 and 9 post-infection (PI) females were dissected in drops of saline solution. In total we dissected 42 females fed on sugar only, 13 females fed on sugar with 5 mM EGTA and 41 females fed on sugar with 20 mM EGTA. The individual guts were checked for presence and localization of Leishmania promastigotes under the light and fluorescence microscope Olympus BX51. Special emphasis was given to the colonization of the stomodeal valve. Levels of Leishmania infections were graded into four categories: negative, light (< 100 parasites/gut), moderate (100-1000 parasites/gut) and heavy (>1000 parasites/gut).

2) Apart from sharing structural elements of the attachment plaque, there is no evidence that the forms that attach in vitro represent an authentic and unique life cycle stage in vivo. If by "haptomonad", the authors are referring to any promastigote that attaches to a surface via the formation of a flagellar attachment plaque, then this should be explained and clarified in the manuscript.

A number of the issues raised by the reviewers focus on the terminology used in this study. The in vitro cultured cells we use to initiate this process are widely regarded as promastigotes. During the process these cells clearly differentiate changing shape and morphology, forming an attachment plaque. in vivo cells exhibiting an attachment plaque were defined as haptomonads (Killick-Kendrick et al., 1974). In the intervening time since this term was coined, based solely on morphology and the formation of an attachment plaque, no molecular markers of haptomonads have been defined. Moreover, other aspects of the biology of these forms is not helpful as haptomonads in other species can occur in other places (e.g. *L. braziliensis* complex in the hindgut; Killick-Kendrick et al., 1977); therefore, the unifying feature of haptomonads is their morphology and the formation of an attachment plaque. It is for that reason we termed them haptomonad. Given the referees uncertainty and lack of molecular markers we have referred to them throughout the manuscript as in vitro haptomonad-like cells. We have revised the manuscript and added explanations to clarify the definition of the in vitro haptomonad-like cell (Line 84-89 and 509-511).

Reviewer #1 (Recommendations for the authors):While the studies lend impressive details to a previously described aspect of promastigote colonization of the stomodeal valve, they do not further our understanding of the consequences of this colonization, if any, to the ability of infected flies to transmit infection by bite. The degeneration of the valve tissue, including a detachment of the cuticle lining that was previously described in infected flies (Schlein et al., PNAS, 1992), was not observed in the current studies. As damage to the valve, mediated by a parasite-derived chitinase, was proposed to facilitate transmission by bite, some discussion as to why they fail to see any damage is warranted.

We thank the reviewer for this comment. In the paper (Schlein et al., 1992) and our previous paper (Volf et al., 2004), a higher infection dose or a longer duration between blood meal and imaging was necessary to see the clear degeneration of the valve tissue. However, based on the reviewer’s comment, we revisited the SBF-SEM data of the stomodeal valve in detail and found clear detachment of the cuticle lining in the areas where a large number of parasites were attached. We have now included the data as “Figure1—figure supplement 3” and added a description (Line 187-190).

Is 'haptomonad' a term used to describe any promastigote seen in attachment to the stomodeal valve, regardless of morphology? The attached, yellow-colorized cell shown in Figures1B and F with an elongated cell body and extended axoneme has the appearance of a metacyclic form. Would not large numbers of metacyclic promastigotes be expected to colonize the anterior midgut in close proximity to the valve? This raises the question as to whether there is a dedicated stage that is pre-adapted to attachment. This could be addressed in part using the in vitro system. Will metacyclics purified from culture attach in vitro, and do they remodel their flagellum and assume a haptomonad morphology?

In this paper, cells are classified as ‘haptomonads’ if they have (i) a short, wide cell body, (ii) a shortened axoneme, and (iii) attached to the substrate through an electron-dense attachment plaque. We regard other attached cells with an elongated flagellum and axoneme with a small attachment plaque (e.g., yellow-colorized cell shown in Figure 1F) as promastigote cells either differentiating to or away from the haptomonad form.

According to the morphological criteria in the previous studies (Rogers et al., 2002), the cells we saw either attached to or near the stomodeal valve had either a metacyclic (Body length <8 μm, body width <1 μm, flagellum > body length) or leptomonad (Body length 6.5 – 11.5 μm, width variable, flagellum > body length) morphology (Figure 1E). In fact the majority of the cells attached to the valve laterally through the flagellum side or were unattached had a leptomonad morphology.

To address whether metacyclics are able to attach, we purified metacyclics *Leishmania* culture as described in Späth and Beverley, Exp. Parasitol., 2001, and conducted the in vitro attachment assay. From this experiment, we confirmed that in vitro purified metacyclics had little ability to attach. We have added this data as “Figure 4—figure supplement 3” and descriptions (Line 399-449 and 628-681). The lack of attachment by metacyclic forms suggests those attached cells with a long flagellum are likely leptomonad forms in the process of attachment, which is consistent with the suggestion of Rogers et al., 2002 who described the leptomonad as the precursor of the haptomonad.

The calcium2+ dependency of the attachment was confined to the in vitro model. Can the divalent cations be chelated in vivo, perhaps by including EGTA in the sugar meals to address this important question in infected sandflies?

We appreciate this suggestion; however, as outlined above, we believe the results of this experiment are difficult to interpret and when we fed EGTA to sand flies there was no effect on haptomonad attachment (Author response images 1-3). We believe EGTA is more suited for use in relatively well defined in vitro conditions such as those we used to examine its effect on cell adhesion not in complex biological systems.

Reviewer #2 (Recommendations for the authors):Line 29-30: haptomonad is indeed a possible critical form for transmission, but it lacks formal demonstration still in all literature available.

We have revised the text referring to the relationship between transmission and the haptomonad form to clarify that a definitive link between the two has not been demonstrated (Line 22-23, 31 and 113-114).

Lines 74-75: The literature cited here by the authors provides an overview of promastigotes attachment and flagellar modifications inside the sand fly vector. At Wakid and Bates, 2004, the different surfaces for in vitro attachment were proposed but no haptomonad was specifically implicated or status of differentiation was proposed and only "promastigotes" were termed. It does show the formation of hemidesmossomes in vitro but this is no guarantee that any promastigote can't do it or isn't an artifact from in vitro conditions. Due to that, authors cannot claim with the data provided that it's a protocol for having haptomonads in vitro. The manuscript should be adjusted to harbor accurate discussion and conclusions. How do the authors guarantee the hemidesmosomes found in vitro are comparable to haptomonads in vivo ones?

We acknowledge that our approach is unable to confirm at the molecular level that the haptomonad-like cells we generate in vitro are equivalent to those observed in the sand fly. As highlighted by the reviewer we do show that the ultrastructural composition of the attachment plaque complex is almost identical between haptomonads in the sand fly and our in vitro haptomonad-like cells. In addition, other aspects of these cells are comparable such as (i) formation of filamentous structures within the flagellum, (ii) shortened axoneme with central pair microtubules, (iii) disassembled PFR, (iv) expanded FAZ structure, (v) cell body dimensions, and (iv) the formation of the extracellular filamentous gel-like matrix. We consider that these aspects support that the in vitro haptomonad-like cells mimic the haptomonads in vivo.

Moreover, given the similarities at the ultrastructural level it would be surprising to find that the proteins within the hemidesmosomes in the sand fly haptomonads differed significantly from those in the in vitro derived haptomonad-like cells and would suggest that the cell can form the same structure in two different ways. However, as the reviewer mentioned, these structural and morphological similarities are not sufficient to ensure that the in vitro haptomonad-like cells are the same as in vivo haptomonads. Therefore, following the reviewer’s suggestion, we have adjusted the manuscript explicitly highlighting the caveats of our approach (Line 84-89, 509-511).

Line 87: Although haptomonads are predicted to have an important role in transmission, no data is available demonstrating its role. Too much to conclude here unless a formal demonstration is provided.

We revised the text at this point as explained above (Line 113-114).

Line 138: how do the authors know the valve is heavily infected as no counts or infection status were provided?

We have vast experience in observing heavily infected sand fly gut and stomodeal valve, and quantitatively assessing the intensity of their infection by counting the infected parasites, as we showed in our previous paper (Sunter et al., 2019). In this instance, the density of the haptomonads in the SBF-SEM slices (Figure 1—video 1 and Figure 1A) is comparable to the light micrographs of the heavily infected stomodeal valve in the previous work (Figure 1 in Rogers et al., 2008). Given the density of the attached haptomonads, we consider that the stomodeal valves used in this study are heavily infected (Figure 1—video 1 and Figure 1A). We have modified the manuscript to explain this (Line 194-196).

Lines 145-147: Can this be due to sample prep/processing? The asymmetry?

We have previously shown that the anterior cell tip in the promastigote cell has an asymmetric extension along one side of the flagellum (Sunter et al., 2019) and can be seen in other papers as well (e.g. Wheeler et al., 2016).This asymmetry results in the anterior end of the cell body crossing the flagellum at an oblique angle when viewed in certain orientations (Figure 1 in Sunter et al., 2019). Given that this feature has been observed independently, it is unlikely an artefact due to sample preparation and/or processing. We consider this to be an intrinsic cell morphological feature of free-swimming promastigotes, and this asymmetric extension of the anterior cell body was much shorter in haptomonads (Figure 2A).

Line 191: Vide comment for lines 74-75.

We have revised the text as mentioned above (Line 264).

Lines 196-200: How the authors are sure of the structure being a clump of haptomonads? And not just an aggregate of cells in multiplication as normally found in cultures?

We believe the large clumps of cells observed by SEM attached to the substrate are a mixture of haptomonad-like cells actually attached to the substrate and promastigotes that are not attached to the substrate, which we have now explicitly outlined in the manuscript. The origins of these clumps are not clear –

They might result from the continued replication of an attached in vitro haptomonad-like cell, which is able to generate a promastigote daughter cell which remains associated with the attached cell, resulting in a clump after several cell cycles.

Or they might result from the attachment of a number of cells within an aggregate found within normal culture.

However, whatever their origin when we examine such a cell clump by SBF-SEM as shown in Figure 3B and Figure 3—figure supplement 1, we observed that these clumps contain many haptomonad-like cells with clear attachment plaques.

Line 227: The discussion on this topic should be carefully revised. For example, video 6 shows an in vitro promastigote attaching to the surface. This attachment has a hemidesmosome structure formation that resembles the in vivo haptomonad ones. However, this is no evidence that haptomonad differentiation is happening in vitro.

We have revised the text describing this experiment (Line 320-386).

Lines 259-269: This section of the manuscript gives interesting information on the location of RSP4/6 and PFR2 on the attaching promastigotes to the scratched surface. Again, it is not conclusive of a haptomonad differentiation situation. The lack of PRF signal can only be concluded by the data presented as "disassembled during promastigote attachment and hemidesmosome formation". This is important information and I suggest the authors explore the strength of the data.

We thank the reviewer for the comment. We have revised the manuscript to make our description more accurate (Line 387-398). We have analysed these data further by measuring the length of the mNeonGreen::PFR2 signal and mCherry::RSP4/6 signal in in vitro adhering cells (Figure 4—figure supplement 2). This showed that 30% of mature in vitro haptomonad-like cells (9/30 cells) appeared to lose the mNG::PFR2 signal, but the mCh::RSP4/6 signal was still present in all the cells.

Lines 271-291: Same issue as before. To claim in vitro haptomonad formation the authors will need a more detailed comparison with fly-derived haptomonads. This is extremely challenging to achieve though. However, the lack of material does not allow for in vitro assumptions claimed in the manuscript. The current manuscript will be significantly more accurate if the authors focus only on the data and interesting details about the flagellum attachment to the surface that "can" be similar to the real scenario inside the sand fly.

We thank the reviewer for the important suggestion. As mentioned above, we have adjusted the text in which describe the relationship between the in vitro haptomonad-like cell and the in vivo haptomonad (Line 84-89, 509-511).

The discussion that follows from here should be adjusted based on the comments above. The data is important and provide details not explored. Claiming it's a haptomonad process in vitro is only damaging the dataset as this cannot be concluded. Also, this manuscript will benefit a lot by having a detailed description of how to analyze and get to the 3D models presented. No detailed description was provided. This had a strong potential for usage beyond the Leishmania/sand fly field.

We have revised the manuscript based on the reviewer’s suggestions to ensure the caveats of our approach are clear (Line 84-89, 509-511). We have included a detailed description of how to analyse the 3D models (Line 756-763), and have added videos showing a rotated view of each 3D model (Figure 1—video 3 and 4, Figure 2—video 2, and Figure 3—video 2 and 4). In addition, we have deposited the SBF-SEM and tomography data on the EMPIAR, so that everyone can access the data (Line 763-766).

Line 478: Haptomonad is a cryptic life stage with an unknown ontology.How can the authors guarantee the in vitro cells have a similar differentiation pattern to haptomonads found in vivo?

As the reviewer suggests, we still need a formal demonstration to guarantee that the in vitro cells have a similar differentiation pattern to in vivo haptomonads. An important step to guarantee this would be to identify molecular markers of haptomonads in the sand fly. However, we agree with the reviewer that, without molecular markers, these data of in vitro haptomonad-like cells adhering may not closely align to the differentiation pattern of haptomonads inside the sand fly. We therefore revised the descriptions in relation to this (Line 84-89, 509-511).

Line 487: Any media formulation changes will likely promote drastic changes on parasite growth and differentiation. Why for in vitro live cell imaging the authors changed the culture media conditions?

We routinely wash cells in DMEM without phenol red and FCS in preparation for live cell light microscopy (e.g. Figure 4D), as the FCS stops the cells becoming immobilised on the glass surface and the phenol red causes an increase the level of background fluorescence in the red channel. DMEM has been used to wash *Leishmania* in previous studies (De Pablos et al., 2019; Hassani et al., 2011), and we ourselves have confirmed that DMEM did not affect cell morphology or protein localisation in short-time observations. However, as we indicated in Materials and methods, in the time-lapse observation, we washed cells with fresh M199 medium and the differentiation of cells was observed in M199 medium. We have adjusted the text to make this clearer (Line 777-778).

Why no statistics detailed methods part is available?

We have now added a statistics section to the Materials and methods (Line 864-868).